# PLANNING WITH SEQUENCE MODELS THROUGH ITERATIVE ENERGY MINIMIZATION

**Hongyi Chen**[*][†]**, Yilun Du** [*][‡]**, Yiye Chen**[*][†]**, Joshua Tenenbaum**[‡]**, Patricio Vela**[†]
Georgia Institute of Technology[†]     Massachusetts Institute of Technology[‡]

## ABSTRACT

Recent works have shown that sequence modeling can be effectively used to train reinforcement learning (RL) policies. However, the success of applying existing sequence models to planning, in which we wish to obtain a trajectory of actions to reach some goal, is less straightforward. The typical autoregressive generation procedures of sequence models preclude sequential refinement of earlier steps, which limits the effectiveness of a predicted plan. In this paper, we suggest an approach towards integrating planning with sequence models based on the idea of iterative energy minimization, and illustrate how such a procedure leads to improved RL performance across different tasks. We train a masked language model to capture an implicit energy function over trajectories of actions, and formulate planning as finding a trajectory of actions with minimum energy. We illustrate how this procedure enables improved performance over recent approaches across BabyAI and Atari environments. We further demonstrate unique benefits of our iterative optimization procedure, involving new task generalization, test-time constraints adaptation, and the ability to compose plans together. Project website: https://hychen-naza.github.io/projects/LEAP.

## 1   INTRODUCTION

Sequence modeling has emerged as unified paradigm to study numerous domains such as language (Brown et al., 2020; Radford et al., 2018) and vision (Yu et al., 2022; Dosovitskiy et al., 2020). Recently, (Chen et al., 2021; Janner et al., 2021) have shown how a similar approach can be effectively applied to decision making, by predicting the next action to take. However, in many decision making domains, it is sub-optimal to simply predict the next action to execute – as such an action may be only locally optimal and lead to global dead-end. Instead, it is more desirable to plan a sequence of actions towards a final goal, and choose the action most optimal for the final overall goal.

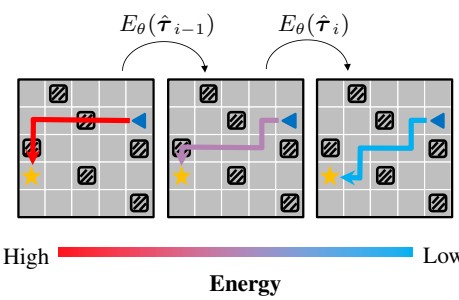

Figure 1: **Plan Generation through Iteratively Energy Minimization.** LEAP plans a trajectory to a goal (specified by the yellow star) by iteratively sampling and minimizing a trajectory energy function estimated using language model $E_\theta$.

Unlike greedily picking the next action to execute, effectively constructing an action sequence towards a given goal requires a careful, iterative procedure, where we need to assess and refine intermediate actions in a plan to ensure we reach the final goal. To refine an action at a particular timestep in a plan, we must reconsider both actions both before and after the chosen action. Directly applying this procedure to standard language generation is difficult, as the standard autoregressive decoding procedure prevents regeneration of previous actions based of future ones. For example, if the first five predicted actions places an agent at a location too far to reach a given goal, there is no manner we may change the early portions of plan.

In this paper, we propose an approach to iteratively generate plans using sequence models. Our approach, M**ul**tistep **E**nergy-Minimiz**a**tion **P**lanner (LEAP), formulates planning as an iterative op-

---

* denotes equal contribution. Correspondence to `hchen657@gatech.edu`, `yilundu@mit.edu`, `yychen2019@gatech.edu`

timization procedure on an energy function over trajectories defined implicitly by a sequence model (illustrated in Figure 1). To define an energy function across trajectories, we train a bidirectional sequence model using a masked-language modeling (MLM) objective (Devlin et al., 2019). We define the energy of a trajectory as the negative pseudo-likelihood (PLL) of this MLM (Salazar et al., 2019) and sequentially minimize this energy value by replacing actions at different timepoints in the trajectory with the marginal estimates given by the MLM. Since our MLM is bi-directional in nature, the choice of new action at a given time-step is generated based on both future and past actions.

By iteratively generating actions through planning, we illustrate how our proposed framework outperforms prior methods in both BabyAI (Chevalier-Boisvert et al., 2019) and Atari (Bellemare et al., 2013) tasks. Furthermore, by formulating the action generation process as an iterative energy minimization procedure, we illustrate how this enables us to generalize to environments with new sets of test-time constraints as well as more complex planning problems. Finally, we demonstrate how such an energy minimization procedure enables us to compose planning procedures in different models together, enabling the construction of plan which achieves multiple objectives.

Concretely, in this paper, we contribute the following: First, we introduce LEAP, a framework through which we may iteratively plan with sequence models. Second, we illustrate how such a planning framework can be beneficial on both BabyAI and Atari domains. Finally, we illustrate how iteratively planning through energy minimization gives a set of unique properties, enabling better test time performance on more complex environments and environments with new test-time obstacles, and the ability to compose multiple learned models together, to jointly generate plans that satisfy multiple sets of goals.

## 2   RELATED WORK

**Sequence Models and Reinforcement Learning.**   Sequence modeling with deep networks, from sequence-to-sequence models (Hochreiter & Schmidhuber, 1997; Sutskever et al., 2014) to BERT (Devlin et al., 2019) and XLnet (Yang et al., 2019), have shown promising results in a series of language modeling problems (Dai et al., 2019b; Sutskever et al., 2014; Liu & Lapata, 2019; Dehghani et al., 2018). With these advances, people start applying sequence models to represent components in standard RL such as policies, value functions, and models to improved performance (Espeholt et al., 2018; Parisotto et al., 2020; Kapturowski et al., 2018). While the sequence models provide memory information to make the agent predictions temporally and spatially coherent, they still rely on standard RL algorithm to fit value functions or compute policy gradients. Furthermore, recent works replace as much of the RL pipeline as possible with sequence modeling to leverage its scalability, flexible representations and causally reasoning (Janner et al., 2021; Chen et al., 2021; Furuta et al., 2021; Zheng et al., 2022; Emmons et al., 2021; Li et al., 2022). However, those methods adopt autoregressive modeling objectives and the predicted trajectory sequences have no easy way to be optimized, which will inevitably lower the long-horizon accuracy. Recent studies point out that using sequence models (Chen et al., 2021; Emmons et al., 2021) rather than typical value-based approaches have difficulty converging in stochastic environments (Paster et al., 2020; 2022).

**Planning in Reinforcement Learning.**   Planning has been explored extensively in model-based RL, which learns how the environment respond to actions (Sutton, 1991). The learned world dynamic model is exploited to predict the conditional distribution over the immediate next state or autoregressively reason the long-term future (Chiappa et al., 2017; Ke et al., 2018). However, due to error compounding, plans generated by this procedure often look more like adversarial examples than optimal trajectories when the planning horizon is extended (Bengio et al., 2015; Talvitie, 2014; Asadi et al., 2018). To avoid the aforementioned issues, simple gradient-free method like Monte Carlo tree search (Coulom, 2007), random shooting (Nagabandi et al., 2018) and beam search (Sun et al., 2022) are explored. Another line of works studied how to break the barrier between model learning and planning, and plan with an imperfect model, include training an autoregressive latent-space model to predict values for abstract future states (Tamar et al., 2016; Oh et al., 2017; Schrittwieser et al., 2020; Sun et al., 2022); energy-based models of policies for model-free reinforcement learning (Haarnoja et al., 2017); improve the offline policies by planning with learned models for model-based reinforcement learning (Yu et al., 2020; Schrittwieser et al., 2021); directly applying collocation techniques for direct trajectory optimization (Erez & Todorov, 2012; Du et al., 2019); and folding planning into the generative modeling process (Janner et al., 2021). In contrast to these works, we explore having planning directly integrated in a language modeling framework.

**Energy Based Learning.** Energy-Based Models (EBMs) capture dependencies between variables by associating a scalar energy to each configuration of the variables, and provide a unified theoretical framework for many probabilistic and non-probabilistic approaches to learning (LeCun et al., 2006). Prior works have explored EBMs for policy training in model-free RL (Haarnoja et al., 2017), modeling the environment dynamics in model-based RL (Du et al., 2019; Janner et al., 2022) and natural images (Du & Mordatch, 2019; Dai et al., 2019a), as well as energy values over text (Goyal et al., 2021). Most similar to our work, Du et al. (2019) illustrates how energy optimization in EBMs naturally support planning given start and goal state distributions. However, the underlying training relies on a constrastive divergence, which is difficult to train (Du et al., 2020). On the other hand, our training approach relies on a more stable masked language modeling objective.

## 3 METHOD

In this section, we describe our framework, **Mu**ltistep **E**nergy-Minimiz**a**tion **P**lanner (LEAP), which formulates planning as a energy minimization procedure. Given a set of trajectories in a discrete action space, with each trajectory containing state and action sequences $(\mathbf{s}_1, \mathbf{a}_1, \mathbf{s}_2, \mathbf{a}_2, \ldots, \mathbf{s}_N, \mathbf{a}_N)$, our goal is to learn a planning model which can predict a sequence of actions $\mathbf{a}_{1:T}$, given the trajectory context $\boldsymbol{\tau}_{ctx}$ containing the past $K$ steps states and actions, that maximizes the long-term task-dependent objective $\mathcal{J}$:

$$\mathbf{a}_{1:T}^* = \arg\max_{\mathbf{a}_{1:T}} \mathcal{J}(\boldsymbol{\tau}_{ctx}, \mathbf{a}_{1:T})$$

where $N$ denotes the length of the entire trajectory and $T$ is the planning horizon. We use the abbreviation $\mathcal{J}(\boldsymbol{\tau})$, where $\boldsymbol{\tau} =: (\boldsymbol{\tau}_{ctx}, \mathbf{a}_{1:T})$, to denote the objective value of that trajectory. To formulate this planning procedure, we learn an energy function $E_\theta(\boldsymbol{\tau})$, which maps each trajectory $\boldsymbol{\tau}$ to a scalar valued energy so that

$$\mathbf{a}_{1:T}^* = \arg\min_{\mathbf{a}_{1:T}} E_\theta(\boldsymbol{\tau}_{ctx}, \mathbf{a}_{1:T}).$$

### 3.1 LEARNING TRAJECTORY LEVEL ENERGY FUNCTIONS

We wish to construct an energy function $E_\theta(\boldsymbol{\tau}_{ctx}, \mathbf{a}_{1:T})$ such that minimal energy is assigned to optimal set actions $\mathbf{a}_{1:T}^*$. To train our energy function, we assume access to dataset of $M$ near optimal set of demonstrations in the environment, and train our energy function to have low energy across demonstrations. Below, we introduce masked language models, and then discuss how we may get such a desired energy function from masked language modeling.

**Masked Language Models.** Given a trajectory of the form $(\mathbf{s}_1, \mathbf{a}_1, \mathbf{s}_2, \mathbf{a}_2, \ldots, \mathbf{s}_n, \mathbf{a}_n)$, we train a transformer language model to model the marginal likelihood $p_\theta(\mathbf{a}_t | \boldsymbol{\tau}_{ctx}, \mathbf{a}_{-t})$, where we utilize $\mathbf{a}_{-t}$ as shorthand for actions $\mathbf{a}_{1:T}$ except the action at timestep $t$. To train this masked language model (MLM), we utilize the standard BERT training objective (Devlin et al., 2019), where we minimize the loss function

$$\mathcal{L}_{\text{MLM}} = \mathbb{E}_{\boldsymbol{\tau}, t} \left[ -\log p_\theta(\mathbf{a}_t; \boldsymbol{\tau}_{ctx}, \mathbf{a}_{-t}) \right], \tag{1}$$

where we mask out and predict the marginal likelihood of percentage of the actions in a trajectory (details on masking in the Section A.4). .

**Constructing Trajectory Level Energy Functions.** Given a trained MLM, we define an energy function for a trajectory as the sum of negative marginal likelihood of each action in a sequence

$$E_\theta(\boldsymbol{\tau}) = -\sum_t \log p_\theta(\mathbf{a}_t; \boldsymbol{\tau}_{ctx}, \mathbf{a}_{-t}). \tag{2}$$

Such an energy function, also known as the pseudolikelihood of the MLM, has been used extensively in prior work in NLP (Goyal et al., 2021; Salazar et al., 2020), and has been demonstrated to effectively score the quality natural language text (outperforming direct autoregressive scoring) (Salazar et al., 2020). In our planning context, this translates to effectively assigning low energy to optimal planned actions, which is our desired goal for $E_\theta(\boldsymbol{\tau})$. We illustrate the energy computation process in Figure 2.

### 3.2 PLANNING AS ENERGY MINIMIZATION

Given a learned energy function $E_\theta(\boldsymbol{\tau})$, which assigns low energy to optimal trajectories $\mathbf{a}_{1:T}^*$, we wish to plan a sequence of actions in test-time to minimize our energy function. To implement this

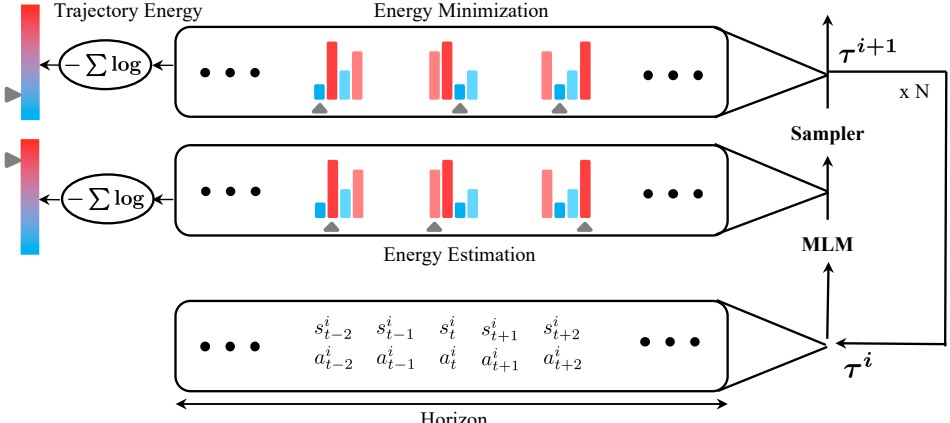

Figure 2: **Energy Minimization in LEAP.** LEAP generates plans via Gibbs sampling different actions based on a learned trajectory energy model $E_\theta(\boldsymbol{\tau})$. In each iteration, Masked Language Model (MLM) predicts the energy of alternative actions at selected timesteps in a trajectory. A new trajectory is generated using a Gibbs sampler, with individual actions sampled based on the energy distribution. By repeating the above steps iteratively, LEAP generates the trajectory with low energy value.

planning procedure, we utilize Gibbs sampling of individual actions at each timestep on the energy function $E_\theta(\boldsymbol{\tau})$ for low energy plans, which we detail below.

For test-time plan generation, we initialize a masked trajectory of length $T$, with a small context length of past states and actions, which we illustrate in Equation 3. At each step of Gibb's sampling, we randomly mask out one or multiple action tokens in padded locations and perform forward pass to estimate their energy distribution conditioned on trajectory context $\boldsymbol{\tau}_{\backslash I}^i$ , which is the masked outcome from previous iteration on sampled timesteps collected in index set $I$. Then, action $\mathbf{a}_t$ is sampled using Gibb's sampling based on the locally normalized energy score $\mathbf{a}_t \sim p_\theta(\mathbf{a}_t; \boldsymbol{\tau}_{ctx}, \mathbf{a}_{-t})$. The process is illustrated in Figure 2, where the actions with low energy values (in blue) are sampled to minimize $E_\theta(\boldsymbol{\tau}^i; \theta)$ in each iteration. To sample effectively from the underlying energy distribution, we repeat this procedure for multiple timesteps, which we illustrate in Algorithm 1. The computational time of Algorithm 1 increases linearly with iteration numbers.

$$\boldsymbol{\tau} = \begin{bmatrix} \mathbf{s}_1 & \mathbf{s}_2 & \dots & \mathbf{s}_{n-1} & \mathbf{s}_n & \mathbf{s}_n & \dots & \mathbf{s}_n \\ \mathbf{a}_1 & \mathbf{a}_2 & \dots & \mathbf{a}_{n-1} & [\mathrm{PAD}] & [\mathrm{PAD}] & \dots & [\mathrm{PAD}] \end{bmatrix} \qquad (3)$$

$$\underbrace{\phantom{\mathbf{s}_1 \quad \mathbf{s}_2 \quad \dots \quad \mathbf{s}_{n-1}}}_{\text{context}} \quad \underbrace{\phantom{\mathbf{s}_n \quad \mathbf{s}_n \quad \dots \quad \mathbf{s}_n}}_{\text{plan}}$$

---

**Algorithm 1** Iterative Planning through Energy Minimization (for discrete actions)

---

1: **Require** trained energy model $h_\theta$, context trajectory $\boldsymbol{\tau}$
2: Pad the states and actions with length $T$ into context trajectory
3: **for** $i = 1, \dots, N$ **do**
4:     // Sample index set.
5:     $I \sim [1, 2, \cdots, T]$
6:     // Estimate the energy distributions on masked tokens
7:     $\mathcal{E} \leftarrow f(h(\boldsymbol{\tau}_{\backslash I}^i; \theta))$
8:     // Sample the action tokens based on energy value
9:     $\mathbf{a} \sim \mathcal{E}$
10:     // Update actions $\mathbf{a}$ in $\boldsymbol{\tau}$ at masked tokens
11:     $\boldsymbol{\tau}^{i+1} \leftarrow \boldsymbol{\tau}_{\backslash T}^i + \mathbf{a}$
12: **end for**
13: Execute all planned actions $\mathbf{a}_{1:T}$ or the first planned action $\mathbf{a}_1$ in padded trajectory $\boldsymbol{\tau}$

---

Note that our resultant algorithm has several important differences compared to sequential model based approaches such as Decision Transformer (DT) (Chen et al., 2021). First, actions are generated using an energy function defined globally across actions, enabling us to choose each action factoring in the entire trajectory context. Second, our action generation procedure is iterative in nature, allowing us to leverage computational time to find better solutions towards final goals.

## 4 PROPERTIES OF MULTISTEP ENERGY-MINIMIZATION PLANNER

In LEAP, we formulate planning as an optimization procedure $\arg\min_{\boldsymbol{\tau}} E_\theta(\boldsymbol{\tau})$. By formulating planning in such a manner, we illustrate how our approach enables online adaptation to new test-time

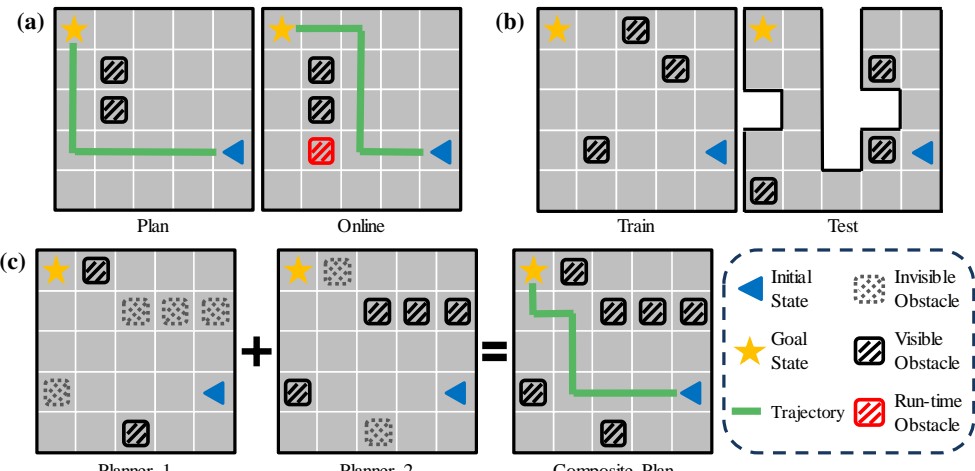

Figure 3: **Properties of Planning as Energy Minimization.** By formulating planning as energy minimization, LEAP enables the following properties: (a): Online adaptation; (b): Generalization; (c): Task composition.

constraints, generalization to harder planning problems, and plan composition to achieve multiple set of goals (illustrated in Figure 3).

**Online adaptation.**    In LEAP, recall that plans are generated by minimizing an energy function $E_\theta(\boldsymbol{\tau})$ across trajectories. At test time, if we have new external constraints, we maybe correspondingly define a new energy function $E_{\text{constraint}}(\boldsymbol{\tau})$ to encode these constraints. For instance, if a state becomes dangerous at test time (illustrated as a red grid in Figure 3 (a)) – we may directly define an energy function which assigns 0 energy to plans which do not utilize this state and high energy to plans which utilize such a state. We may then generate plans which satisfies this constraint by simply minimizing the summed energy function

$$\boldsymbol{\tau}^* = \arg\min_{\boldsymbol{\tau}}(E_\theta(\boldsymbol{\tau}) + E_{\text{constraint}}(\boldsymbol{\tau})). \qquad (4)$$

and then utilize Gibb's sampling to obtain a plan $\boldsymbol{\tau}^*$ from $E_\theta(\boldsymbol{\tau}) + E_{\text{constraint}}(\boldsymbol{\tau})$. While online adaptation may also be integrated with other sampling based planners using rejection sampling, our approach directly integrates trajectory generation with online constraints.

**Novel Environment Generalization.**    In LEAP, we leverage many steps of sampling to recover an optimal trajectory $\boldsymbol{\tau}^*$ which minimizes our learned trajectory energy function $E_\theta(\boldsymbol{\tau})$. In settings at test time when the environment is more complex than those seen at training time (illustrated in Figure 3 (b)) – the underlying energy function necessary to compute plan feasibility may remain simple (i.e. measure if actions enter obstacles), but the underlying planning problem becomes much more difficult. In these settings, as long as the learned function $E_\theta(\boldsymbol{\tau})$ generalizes, and we may simply leverage more steps of sampling to recover a successful plan in this more complex environment.

**Task compositionality.**    Given two different instances of LEAP, $E_\theta^1(\boldsymbol{\tau})$ and $E_\theta^2(\boldsymbol{\tau})$, encoding separate tasks for planning, we may generate a trajectory which accomplishes the tasks encoded by both models by simply minimizing a composed energy function (assuming task independence)

$$\boldsymbol{\tau}^* = \arg\min_{\boldsymbol{\tau}}(E_\theta^1(\boldsymbol{\tau}) + E_\theta^2(\boldsymbol{\tau})). \qquad (5)$$

An simple instance of such a setting is illustrated in Figure 3 (c), where the first LEAP model $E_\theta^1(\boldsymbol{\tau})$ encodes two obstacles in an environment, and a second LEAP model $E_\theta^2(\boldsymbol{\tau})$ encodes four other obstacles. By jointly optimizing both energy functions (through Gibbs sampling), we may successfully construct a plan which avoids all obstacles in both models.

## 5   EXPERIMENTS

In this section, we evaluate the planning performance of LEAP in BabyAI and Atari environments. We compare with a variety of different offline reinforcement learning approaches, and summarize the main results in Figure 4.

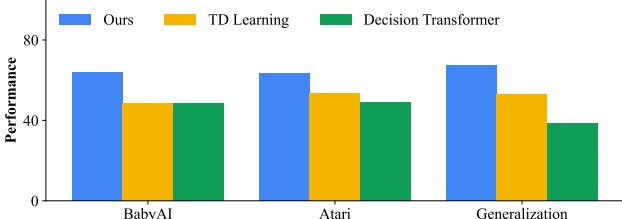

Figure 4: **Quantitative Results of LEAP of Different Domains.** Results comparing LEAP to Decision Transformer and TD learning (IQL in BabyAI and Generalization tests and CQL in Atari) across BabyAI, Atari, and Generalization Tests. On a diverse set of tasks, LEAP performs better than prior approaches.

## 5.1 BABYAI

**Setup** The BabyAI comprises an extensible suite of tasks in environments with different sizes and shapes, where the reward is given only when the task is successfully finished. We evaluate models in trajectory planning that requires the agent move to the goal object through turn left, right and move forward actions, and general instruction completion tasks in which the agent needs to take extra pickup, drop and open actions to complete the instruction. For more detailed experimental settings and parameters, please refer to Table 5 in Appendix.

**Baselines** We compare our approach with a set of different baselines: Behavior Cloning algorithm (BC); model free reinforcement learning (RL) algorithms Batch-Constrained deep Q-Learning (BCQ) (Fujimoto et al., 2019), and Implicit Q-Learning (IQL) (Kostrikov et al., 2021); model based RL algorithm Model-based Offline Policy (MOPO) (Yu et al., 2020); return-conditioning approach Decision Transformer (DT) (Chen et al., 2021); model based Planning Transformer (PlaTe) (Sun et al., 2022). In our experiments, we use, as model inputs, the full observation of the environment, the instruction, the agent's current location and the goal object location (if available).

**Results** Across all environments, LEAP achieves highest success rate, with the margin magnified on larger, harder tasks, see Table 1. In particular, LEAP could solve the easy tasks like GoToLo-calS8N7 with nearly 90% success rate, and has huge advantages over baselines in the large maze worlds (GoToObjMazeS7) and complex tasks (GoToSeqS5R2) which require going to a sequence of objects in correct order. In contrast, baselines perform poorly solving these difficult tasks.

Next, we visualize the underlying iterative planing and execution procedure in GoToLocalS8N7 and GoToSeqS5R2 tasks. On the left side of Figure 5, we present how the trajectory is optimized and its energy is minimized at single time step. Through iterative refinement, the final blue trajectory is closer to the optimal solution than the original red one, which follows the correct direction with higher efficiency and perform the actions like open the door in correct situation. On the right side, we present the entire task completion process through many time steps. LEAP successfully plans a efficient trajectory to visit the two objects and opens the door when blocked. We also explore the model performance in the stochastic settings, please refer to Appendix B.

| Task | Env | BC | BCQ | IQL | DT | PlaTe | MOPO | LEAP |
|------|-----|----|-----|-----|----|-------|------|------|
| Trajectory Planning | GoToLocalS7N5 | 71.0% | 71.5% | 84.5% | 73.0% | 42.5% | 87.0% | **91.0%** |
| | GoToLocalS8N7 | 61.5% | 63.0% | 71.5% | 63.5% | 45.0% | 81.5% | **95.0%** |
| | GoToObjMazeS4 | 24.0% | 23.0% | 52.5% | 46.5% | 35.5% | 60.0% | **62.5%** |
| | GoToObjMazeS7 | 18.0% | 10.5% | 29.0% | 22.0% | 27.5% | 30.5% | **42.5%** |
| Instruction Completion | PickUpLoc | 57.5% | 58.5% | 41.0% | 59.5% | 7.5% | 43.5% | **67.0%** |
| | GoToSeqS5R2 | 13.5% | 10.0% | 28.5% | 26.5% | 25.0% | 30.0% | **33.0%** |
| | GoToObjMazeS4R2Close | 24.0% | 22.5% | 31.5% | 48.5% | 32.5% | 36.0% | **55.0%** |

Table 1: **BabyAI Quantitative Performance.** The task success rate of LEAP and a variety of prior algorithms on BabyAI env. Models are trained with 500 optimal trajectory demos in each environment, and results are averaged over 5 random seeds. The SX and NY in environment name means its size and the number of obstacles.

**Effect of Iterative Refinement.** We investigate the effect of iterative refinement by testing the success rate of our approach under different sample iteration number in GoToLocalS7N5 environment. From the left side of Figure 6, the task success rate continues to improve as we increase the number of sample iteration.

**Energy Landscape Verification.** We further verify our approach by visualizing the energy assignment on various trajectories in the same environment as above. More specifically, we com-

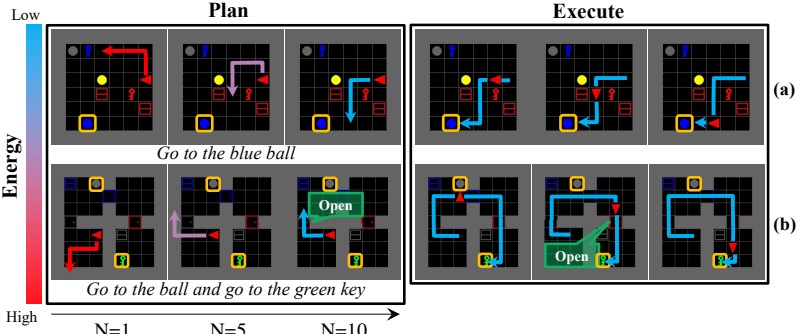

Figure 5: **Qualitative Visualization of Planning and Execution Procedure in BabyAI**. *Left* depicts the planning through iterative energy minimization where $N$ is the sample iteration number. *Right* shows the execution of the concatenate action sequences. Two task settings are illustrated: **(a)**: **Trajectory planning**, where the task is to solely plan a trajectory leading to the goal location. **(b)**: **Instruction completion**, where a sequence of tasks are commanded, and an additional "Open" is involved to get through the doors. Target locations are marked with ☐.

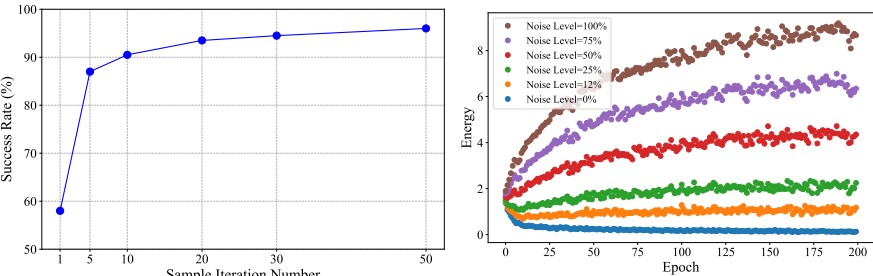

Figure 6: **Analysis of LEAP in the BabyAI Environments**. *Left*: Success rate increases with more sampling steps, suggesting the importance of iterative refinement in LEAP. *Right*: LEAP captures the correct energy landscape. It assigns low energy to the optimal trajectory (noise level=0%) and high energy to noisy paths.

pare the estimated energy of the labeled optimal trajectory with the noisiness of trajectories, produced by randomizing a percentage of steps in the optimal action sequence. The right Figure 6 depicts the energy assignment to trajectories with various corruption levels as a function of training time. With the progress of training, LEAP learns to (a) reduce the energy value assigned to the optimal trajectory; (b) increase the energy value assigned to the corrupted trajectories. This result justifies the performance of LEAP.

**Effect of Training Data.** In BabyAI, we utilize a set of demonstrations generated using an oracle planner. We further investigate the performance of LEAP when the training data is not optimal. To achieve it, we randomly swap the decisions in the optimal demonstration with an arbitrary action with the probability of 25%. We compare against DT, the autoregressive sequential planner. Despite a small performance drop in Table 2, LEAP still substantially outperforms DT, indicating LEAP works well with non-perfect data.

| Env | Optimal | | Suboptimal(25%) | |
|---|---|---|---|---|
| | DT | LEAP | DT | LEAP |
| GoToObjMazeS4 | 46.5% | **65.0%** | 45% | **63.0%** |
| GoToObjMazeS7 | 22.0% | **45.5%** | 19.5% | **44.0%** |
| GoToSeqS5R2 | 26.5% | **38.0%** | 29.5% | **37.5%** |

Table 2: **Performance on Suboptimal Data** The task success rate of LEAP and DT, using optimal trajectories and suboptimal trajectories containing 25% random actions respectively as training data.

### 5.2 ATARI

**Setup** We further evaluate our approach on the dynamically changing Atari environment, with higher-dimensional visual state. Due to above features , we train and test our model without the goal state, and update the plan after each step of execution to avoid unexpected changes in the world. We compare our model to BC, DT (Chen et al., 2021), CQL (Kumar et al., 2020), REM (Agarwal et al., 2020), and QR-DQN (Dabney et al., 2018). Following Chen et al. (2021), the evaluation is conducted on four Atari games (Breakout, Qbert, Pong, and Seaquest), where 1% of data is used for training. Human normalized score is utilized for the performance evaluation.

| Game | LEAP | DT | CQL | QR-DQN | REM | BC |
|---|---|---|---|---|---|---|
| Breakout | **400.9±54.0** | 267.5±56.3 | 211.1 | 17.1 | 8.9 | 138.9 |
| Qbert | 19.5±1.6 | 15.4±6.6 | **104.2** | 0.0 | 0.0 | 17.3 |
| Pong | 108.9±1.6 | 106.1±4.7 | **111.9** | 18.0 | 0.5 | 85.2 |
| Seaquest | 1.3±0.2 | **2.5±0.2** | 1.7 | 0.4 | 0.7 | 2.1 |
| Avg | **127.2** | 97.9 | 107.2 | 8.9 | 2.5 | 60.9 |

Table 3: **Quantitative Comparison on Atari.** Gamer-normalized scores for the 1% DQN-replay Atari dataset (Agarwal et al., 2020). We report the mean and standard error score across 5 seeds. LEAP achieves best averaged scores over 4 games and performs comparably to DT and CQL over all games.

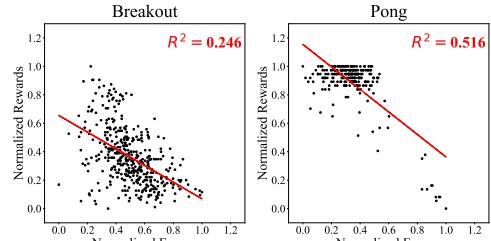

Figure 7: **Energy vs. Reward on Atari.** Energies and rewards are normalized to $[0, 1]$. We demonstrate negative correspondence between the achieved rewards and estimated energy by LEAP, which justifies our method.

**Results** Table 3 shows the result comparison. LEAP achieves the best average performance. Specifically, it achieves better or comparable result in 3 games out of 4, whereas baselines typically perform poorly in more than one games.

**Energy Landscape Verification** In Atari environment, the training trajectories are generated by an online DQN agent during training, whose accumulated rewards are varied a lot. LEAP is trained to estimate the energy values of trajectories depending on their rewards. In Figure 7, we visualize the estimated energy of different training trajectories and their corresponding rewards in Breakout and Pong games. We notice that the underlying energy value estimated to a trajectory is well correlated with its reward, with low energy value assigned to high reward trajectory. This justifies the correctness of our trained model and further gives a natural objective to assess the relative of quality planned trajectory. In Qbert and Seaquest games that LEAP gets low scores, this negative correlation is not obvious showing that the model is not well-trained.

# 6 PROPERTIES OF MULTISTEP ENERGY-MINIMIZATION PLANNER

Next, we analyze the unique properties enabled by LEAP described in §4 in customized BabyAI environments. For each environment, we design at most three settings with increasing difficulty levels to gradually challenge the planning model. As before, the target reaching success rate is measured as the evaluation criteria. The performances are compared against Implicit Q-Learning (IQL) (Kostrikov et al., 2021) and Decision Transformer (DT) (Chen et al., 2021).

## 6.1 ONLINE ADAPTATION

**Setup** To examine LEAP's adaptation ability to test-time constraints, we construct an BabyAI environment with multiple lethal lava zones at the test time as depicted in Figure 8 (a). The planner $E_\theta$ generates the trajectory without the awareness of the lava zones. Once planned, the energy prediction is corrected by the constraint energy function $E_{\text{constraint}}(\tau)$, which assigns large energy to the immediate action leading to lava, and zero otherwise. The agent traverses under the guidance of the updated energy estimation. To make the benchmark comparison fair, we also remove the immediate action that will lead to a lava for all baselines. The difficulty levels are characterized by the amount of lava grids added and the way they are added, where *easy*, *medium* correspond to adding at most 2 and 5 lava grids respectively on the way to the goal object in 8×8 grids world. The third case is hard due to the unstructured maze world in which the narrow paths can be easily blocked by lava grids and requires the agents to plan a trajectory to bypass.

**Results** The quantitative comparison is collected in the Table 4, *Left*. Although drops with harder challenges, the performance of our model still exceeds both baselines under all settings. Visual illustration of *medium* example results can be seen in Figure 8 (a) that the agent goes up first to bypass the lava grids and then drives to the goal object.

## 6.2 NOVEL ENVIRONMENT GENERALIZATION

**Setup** To evaluate LEAP's generalization ability in unseen testing environments, we train the model in easier environments but test them in more challenging environments. In *easy* case, the model is trained in 8×8 world without any obstacles but tested in the world with 14 obstacles as distractors. In *medium* and *hard* cases, the model is trained in single-room world but tested in maze

| Test | Online Adaptation | | | Generalization | | | Task Composition | | |
|---|---|---|---|---|---|---|---|---|---|
| | **LEAP** | DT | IQL | **LEAP** | DT | IQL | **LEAP** | DT | IQL |
| Easy | **92.0**% | 68.0% | 90.5% | **77.5**% | 36.0% | 60.5% | **83.5**% | 58.0% | 42.5% |
| Medium | **64.5**% | 20.0% | 52.0% | **64.0**% | 37.5% | 57.5% | **43.0**% | 15.5% | 11.5% |
| Hard | **54.5**% | 48.0% | 44% | 61.0% | 24.5% | **65.5**% | N/A | N/A | N/A |

Table 4: **Property Test on Modified BabyAI Environments.** Three properties performance of LEAP and prior algorithms on BabyAI tasks with different difficulty. *Left*: Online Adaptation; *Middle*: Generalization; *Right*: Task Composition.

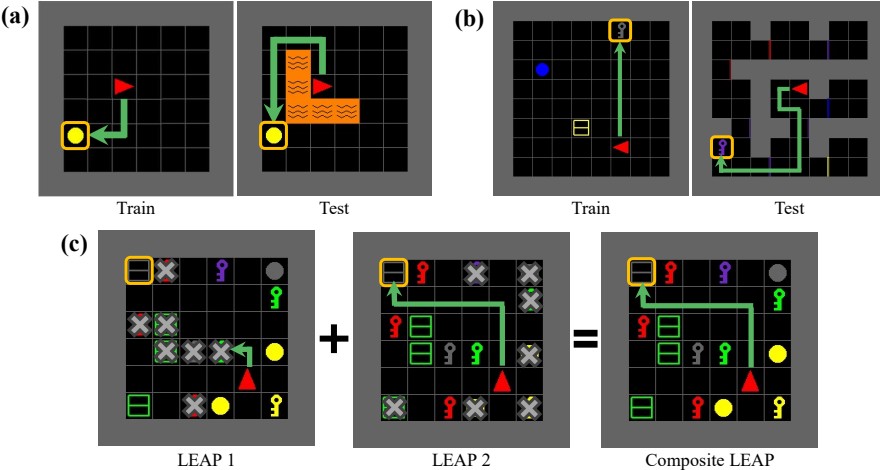

Figure 8: **Qualitative Visualization of Generalization Tests.** (a): Online adaptation (*medium*), trained in plane world and tested in world with lavas; (b): Generalization (*hard*), trained in plane world and tested in maze world; (c): Task composition (*easy*), each model only perceive half number of obstacles. Target locations and unperceivable obstacles are marked with ☐ and ✖, respectively.

world containing multi-rooms connected by narrow paths (Figure 8 (b)). The maze size and the number of rooms in *hard* case are 10×10 and 9, which are larger than 7×7 and 4 in *medium* case.

**Results** Our model achieves best averaged performance across three cases, but slightly worse than IQL in *hard* case, see Table 4, *Middle*. In contrast, sequential RL model DT has significantly lower performance when moved to in unseen maze environments. LEAP trained in plane world could still plan a decent trajectory in unseen maze environment after blocked by walls, see Figure 8 (b).

### 6.3 TASK COMPOSITIONALITY

**Setup** We design composite trajectory planning and instruction completion tasks for *easy* and *medium* cases respectively. In *easy* case, all obstacles are equally separated into two subsets, each observable by one of the two planners, see Figure 8 (c). As a result, the planning needs to add up model's predictions using two partial observations to successfully avoid the obstacles. In *medium* case, two separate models trained for different tasks; one for planning trajectory in 10×10 maze world and the other for object pickup in single-room world. The composite task is to complete the object pickup in 10×10 maze world.

**Results** Our model significantly outperforms the baselines in both two testing cases, while IQL and DT suffer great success rate drop indicating they can not be applied to composite tasks directly, see Table 4, *Right*. This proves that LEAP can be easily combined with other models responsible for different tasks, making it more applicable and general for wide-range tasks. In Figure 8 (c), the composite LEAP could reach the goal by avoiding all obstacles even though the first LEAP planner is blocked by unperceivable obstacle.

## 7 CONCLUSION

This work proposes and evaluates LEAP, an sequence model that plans and refines a trajectory through energy minimization. The energy-minimization is done iteratively - where actions are sequentially along a trajectory. Our current approach is limited to discrete spaces – relaxing this using approaches such as discrete binning (Janner et al., 2021) would be interesting.

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

# Appendix

## A EXPERIMENTAL DETAILS

### A.1 BABYAI ENVIRONMENT DETAILS

We categorize the environments tested in the trajectory planning and instruction completion into the single-room plane world and multi-room maze world which are connect by doors.

1. Trajectory planning:
   - Plane world: GoToLocalS7N5 (7×7), GoToLocalS8N7 (8×8)
   - Maze world: GoToObjMazeS4 (10×10), GoToObjMazeS7 (19×19)
2. Instruction completion:
   - Plane world: PickUpLoc (8×8)
   - Maze world: GoToObjMazeS4R2Close (7×7), GoToSeqS5R2 (9×9)

The Table 5 presents the detailed BabyAI environment settings including the environment size, the number of rooms, the number of obstacles, the status of doors and one example instruction in that environment.

| Env | Size | # Room | # Obs | Door | Inst |
|---|---|---|---|---|---|
| GoToLocalS7N5 | $7 \times 7$ | 1 | 5 | − | go to the green key |
| GoToLocalS8N7 | $8 \times 8$ | 1 | 7 | − | go to the blue box |
| GoToObjMazeS4 | $10 \times 10$ | 9 | 1 | 9 *Open* | go to the blue key |
| GoToObjMazeS7 | $19 \times 19$ | 9 | 1 | 9 *Open* | go to the grey ball |
| GoToObjMazeS4R2Close | $7 \times 7$ | 4 | 1 | 3 *Closed* | go to the blue ball |
| PickUpLoc | $8 \times 8$ | 1 | 8 | − | pick up the yellow ball |
| GoToSeqS5R2 | $9 \times 9$ | 4 | 4 | 3 *Closed* | go to the green door and go to the green door, then go to a red door and go to the green door |

Table 5: BabyAI environment setting details and example instruction.

### A.2 NETWORK DETAILS

We build our LEAP implementation based on Decision Transformer (https://github.com/kzl/decision-transformer) and exploit the instruction encoder from the BabyAI agent model (https://github.com/mila-iqia/babyai/blob/iclr19/babyai/model.py). In detail, we use the Gated Recurrent Units (GRU) encoder to process the instruction and then apply ExpertControllerFiLM (inspired by FiLMedBlock from https://arxiv.org/abs/1709.07871)to fuse the instruction embedding with state embedding. For all our experiments we use bidirectional mask in transformer attention layer, except for Atari where we found casual attention to perform better. The full list of hyperparameters can be found in Table 6 and Table 7, most of the hyperparameters are taken from Decision Transformer and BabyAI agent model.

### A.3 BASELINE MODELS

**BabyAI Baseline Models** We ran BCQ and IQL based on the following implementation

https://github.com/sfujim/BCQ.

https://github.com/BY571/Implicit-Q-Learning/tree/main/discrete_iql.

For BC and DT, we use the author's original implementation

https://github.com/kzl/decision-transformer.

For PlaTe, we use the author's original implementation

| Hyperparameter | Value |
|---|---|
| Number of layers | 3 |
| Number of attention heads | 4 |
| Embedding dimension | 128 |
| Batch size | 64 |
| Image Encoder | nn.Conv2d |
| Image Encoder channels | $128, 128$ |
| Image Encoder filter sizes | $2 \times 2, 3 \times 3$ |
| Image Encoder maxpool strides | $2, 2$ (Image Encoder may vary a little depending on the environment size) |
| Instruction Encoder | nn.GRU |
| Instruction Encoder channels | 128 |
| State Encoder | nn.Linear |
| State Encoder channels | $256, 256, 128$ |
| Max epochs | 200 |
| Dropout | 0.1 |
| Learning rate | $6 * 10^{-4}$ |
| Adam betas | $(0.9, 0.95)$ |
| Grad norm clip | 1.0 |
| Weight decay | 0.1 |
| Learning rate decay | Linear warmup and cosine decay (see code for details) |

Table 6: Hyperparameters of LEAP for BabyAI experiments.

| Hyperparameter | Value |
|---|---|
| Number of layers | 6 |
| Number of attention heads | 8 |
| Embedding dimension | 128 |
| Batch size | 64 Breakout, Qbert |
| | 128 Seaquest |
| | 256 Pong |
| Image Encoder | nn.Conv2d |
| Image Encoder channels | $32, 64, 64$ |
| Image Encoder filter sizes | $8 \times 8, 4 \times 4, 3 \times 3$ |
| Image Encoder strides | $4, 2, 1$ |
| Max epochs | 10 |
| Dropout | 0.1 |
| Learning rate | $6 * 10^{-4}$ |
| Adam betas | $(0.9, 0.95)$ |
| Grad norm clip | 1.0 |
| Weight decay | 0.1 |
| Learning rate decay | Linear warmup and cosine decay (see code for details) |

Table 7: Hyperparameters of LEAP for Atari experiments.

https://github.com/Jiankai-Sun/plate-pytorch.

For MOPO, we use the author's original implementation of dynamic model training and policy learning. For RL policy, we adopt the IQL discussed above.

https://github.com/tianheyu927/mopo.

The actor network and policy network of BCQ and IQL use the transformer architecture which is the same as architecture in our model, see details above. The original DT and BC already use the transformer architecture so we didn't change. For all baselines, we add the same instruction encoder and image encoder described above to process instruction and image observations.

**Atari Baseline Models**    The scores for DT, BC, CQL, QR-DQN, and REM in Table 3 can be found in Chen et al. (2021).

## A.4    Experiment details

**BabyAI**    For LEAP, the larger size environment requires longer horizon $T$ and correspondingly more sampling iterations $N$. After $N$ iteration, all $T$ planned actions will be executed. For DT model, it's beneficial of using a longer context length in more complex environments as shown in its original paper (Chen et al., 2021). We list out these parameters for LEAP and DT models in Table 8. We didn't use context information in LEAP in most BabyAI environments as we expect the iterative planning could generate a correct trajectory based solely on the current state observation. While the GoToSeqS5R2 environment requires go to a sequence of objects in correct order and LEAP needs to remember what objects have been visited from the context. During training, we randomly select and mask one action in a trajectory.

| Env | context | LEAP plan | LEAP sample iteration | DT context |
|---|---|---|---|---|
| GoToLocalS7N5 | 0 | 5 | 10 | 5 |
| GoToLocalS8N7 | 0 | 5 | 10 | 5 |
| GoToObjMazeS4 | 0 | 10 | 30 | 10 |
| GoToObjMazeS7 | 0 | 15 | 50 | 15 |
| GoToObjMazeS4R2Close | 0 | 5 | 10 | 5 |
| PickUpLoc | 0 | 5 | 10 | 5 |
| GoToSeqS5R2 | 20 | 5 | 10 | 20 |

Table 8: BabyAI environment experiment details for LEAP and DT.

The input to DT model includes the instruction, state context sequence, action context sequence and return-to-go sequence in which the target reward is set to 1 initially. The input to other baseline models are the same except they use normal reward sequence instead of return-to-go sequence. While LEAP only use the instruction, state context sequence and action context sequence. Inside state sequence, each state $\mathbf{s}_n$ contains the $[x, y, dir, g_x, g_y]$ meaning the agent's x position, y position, direction and goal object's x position, y position (if the goal location is available).

**Atari**    In dynamically changing Atari environment, LEAP use context information in all four games and only execute the first planned action to avoid the unexpected changes in the world, see details in Table 9. During training, we randomly sample and mask one action in a trajectory.

| Env | context | plan | sample iteration |
|---|---|---|---|
| Breakout | 25 | 5 | 10 |
| Qbert | 25 | 5 | 10 |
| Pong | 25 | 5 | 10 |
| Seaquest | 25 | 10 | 30 |

Table 9: Atari environment experiment details for LEAP.

| Env | w/o return | w return |
|---|---|---|
| Breakout | 182.0 | 378.9 |
| Qbert | 41.0 | 19.6 |
| Pong | 100.7 | 108.9 |
| Seaquest | 0.5 | 1.3 |

Table 10: LEAP performance in Atari environment with and w/o return input.

Note that our approach can easily be conditioned on total reward, by simply concatenating the reward as input in the sequence model. One hypothesis is that when demonstration set contained trajectories of varying quality, taking reward as input following will enable the model to recognize the quality of training trajectories and potentially improve the performance. To further validate the importance of the rewards, we test the LEAP with and without return-to-go inputs, which sum of future rewards Chen et al. (2021). The results show that the performance degradation without the return-to-go inputs, which is shown in Table 10.

## B    Stochastic Environment Testing

In this section we demonstrate the possibility of extending our method into stochastic settings. Although Paster et al. (2022) reveals that planning by sampling from the learnt policy conditioned on desired reward could lead to suboptimal outcome due to the existence of stochastic factors, our

model circumvents the problem by formulating the planning as an optimization problem - we use the Gibbs sampling method to find the trajectory with the lowest energy evaluated by the trained model. Assuming that the frequency of successful actions dominates in the dataset, our model is trained to assign lower energy to trajectories with higher likelihood of reaching the goal. Consequently, in the stochastic environments, LEAP constructs a sequence of actions that has the best opportunity to accomplish the target. When executing this plan in a stochastic environment, we may also choose to replan our sequence of actions after each actual action in the environment (to deal with stochasticity of the next state given an action). Note then that this sequence of actions will be optimal in the stochastic environment, as we always choose the action that has the maximum likelihood of reaching the final state. Also note that multi-step planning can potentially provide advantage over a simple policy to predict the next action in stochastic environments, as such policy simply assigns probability distribution to the immediate next step without the awareness of future step adjustments facing stochastic factors.

To verify the assumptions, we constructed a stochastic testing in BabyAI environment. The test is created by adopting a stochastic dynamic model, where the agent fails to execute the turning actions *turn left/right* with $20\%$ chance, and instead performs the remaining actions, including *turn right/left*, *forward*, *pickup*, *drop*, and *open*, with uniform probability. The remaining settings follow BabyAI experiments detailed in Appendix. A.4, except that we train models using demonstrations generated with the above dynamic model. Those training data are noisy in the sense that the actions taken are not optimal, and corrections are required from future actions. We believe LEAP, as a multi-step planner, can learn the above correlations between the consecutive actions. We compare LEAP with the baseline DT, the results of which is collected in the Table. 11. It can be observed that LEAP has a superior performance compared to DT on both tested environments, which indicates both the possibility of applying our approach in the stochastic settings, and the advantage of multi-step planning when facing stochastic factors.

| Env | LEAP | DT |
|---|---|---|
| GoToObjMazeS4 | **57.5%** | 30.8% |
| GoToObjMazeS7 | **33.3%** | 28.3% |

Table 11: Comparison of LEAP and DT on stochastic settings

## C  ABLATION ON ENERGY MODEL AND OPTIMIZATION METHOD

We further ablate on our design choices, including the energy model and sampling methods. We consider Masked Language Model (MLM) and sequence model classifier as the energy model, and random-shooting, Cross-Entropy Method (CEM), and Gibbs Sampling as the optimization approach. All combinations are tested, for which the results are collected in Table 12. We observe that the Gibbs sampling gives the best outcome with MLM model and that defining an energy value using a sequence model classifier doesn't work well in all settings.

1. Sampling methods:
   - Random-shooting: randomly generated 30 action sequences and pick up the one with lowest estimated energy value.
   - Cross-Entropy method: randomly generated 30 action sequences, keep the three best sequences with lowest estimated energy values in each iteration. Then we randomly update one action token in the elite sequences to get 30 new sequences for next iteration. The sequence with lowest energy is selected in the final iteration.
   - Gibbs sampling: discussed in the main text.

2. Energy models:
   - Sequence model classifier: LSTM sequence model predicts the scalar energy value given the entire trajectory $\tau$, and train the loss between ground truth trajectory energy and estimated energy. The optimal trajectories in Babyai are assigned with lowest energy value 0 and the generates suboptimal trajectories are assigned with higher values depending on the degree of randomness.
   - MLM: discussed in the main text.

| Energy model | Random-shooting | CEM | Gibbs sampling |
|:---:|:---:|:---:|:---:|
| MLM | 23.3% | 57.5% | **62.5%** |
| Classifier | 25.0% | 12.5% | 15.5% |

Table 12: Comparison of different combination between energy models and sampling methods

