# OpenReview forum: "Planning with Sequence Models through Iterative Energy Minimization"
_ICLR.cc/2023/Conference — ICLR 2023 poster_

### Official Review · Reviewer_VX64 · 2022-10-22

**Confidence:** 4
**Correctness:** 3
**Technical Novelty And Significance:** 4
**Empirical Novelty And Significance:** 3
**Recommendation:** 6

**Clarity, Quality, Novelty And Reproducibility:**

* The quality of the paper is in general OK but there seem to be some notation problems.
* The clarity is not good. I'm still a bit confused about a lot of technical details after reading through it.
* The paper is quite original for RL community.
* Code is not provided, due to the low clearance of the methodology, it can be very difficult to reproduce the results even if the hyperparameters are provided.

**Strength And Weaknesses:**

Strength:
* The proposed method is rather novel for RL planning as it goes beyond step-by-step planning which gives extra flexibility and might lead to a new thread of future work.
* Empirical results show LEAP can fit multiple scenarios of planning, eg. goal-oriented, reward maximization and online adaptation by a constraint function.

Weakness:
* Empirical evaluations did not include any other planning-based (actually any model-based) methods. It's then hard to compare the proposed method with conventional planning.
* some of the notations are confusing.
  * The trajectory length is first defined as $n$ and then it redefined as $T$.
  * I'm also not sure what does eq (3) mean. Why all the padded actions correspond to state $s_n$?
  * eq(2) defines energy function as pseudolikelihood but then in the Atari experiment, it seems like a different function is used which is associated with reward. Also, if the optimization objective can be anything, why it's called energy?

Questions:
1. Does $\pmb s$ in eq (1) and eq (2) mean $\pmb{s}_{1}$? So the "model" is not predicting states?
2. Following question 1, if that's the case, how can you define constraint based on future states in eq (4)? Are you using the simulator for planning?
3. How did you get the value and reward estimation for Atari experiments?
4. What did you get the goal state for BABYAI tasks?


**Summary Of The Paper:**

This paper proposed a novel approach (LEAP) for modelling RL trajectories and planning with the model. Unlike GPT-2/Trajectory Transformer style of autoregressive model, it applies BERT style modelling where the actions can be masked in any position of the trajectory. With such a model, planning does not have to follow temporal order. The empirical results show LEAP can do flexible planning and surpass model-free offline RL methods in terms of success rate or return.

**Summary Of The Review:**

I think the idea of the paper is interesting and the empirical studies also show great potential for this work. However, the write-up of the paper still needs to be improved to improve clarity. Current I still have a lot of questions about the method, which may affect my final recommendation but in general, I'm leaning to accept the paper.

---

> ### Author Response · Authors · 2022-11-15
> **Reviewer VX64 Response**
>
> Thank you for your detailed comments. We have added two model-based planning benchmarks and added clarifications about the symbols and method settings.
>
> **Q1) Empirical evaluations about planning-based and model-based methods.**
>
> This was requested by other reviewers as well. We agree that the suggested experiments will add to the paper significantly, and we benchmark two model-based planning baselines PlaTe [1] and MOPO[2] whose results are collected in Table1. Our LEAP model still outperforms these methods in Babyai Env. We have cited PlaTe and MOPO in the related works section of our paper.
>
> | Task | Env | LEAP | PlaTe | MOPO |
> | ------------- | ------------- | ------------- | ------------- | ------------- |
> | TP | GoToLocalS7N5 | **89.5%** | 42.5% | 87.0% |
> | TP | GoToLocalS8N7 | **89.0%** | 45.0% | 81.5% |
> | TP | GoToObjMazeS4 | **65.0%** | 35.5% | 60.0% |
> | TP | GoToObjMazeS7 | **45.5%** | 30.5% | 30.5% |
> | IC | PickUpLoc | **65.0%** | 7.5% | 43.5% |
> | IC | GoToSeqS5R2 | **38.0%** | 25.0% | 30.0% |
> | IC | GoToObjMazeS4R2Close | **55.5%** | 32.5% | 36.0% |
>
> 1. PlaTe: Visually-Grounded Planning with Transformers in Procedural Tasks. Jiankai Sun, De-An Huang, Bo Lu, Yun-Hui Liu, Bolei Zhou, Animesh Garg.
> 2. Mopo: Model-based offline policy optimization. Yu T, Thomas G, Yu L, Ermon S, Zou JY, Levine S, Finn C, Ma T.
>
> **Q2) Notation confusion. Some of the notations are confusing. The trajectory length is first defined as n and then it redefined as T; I'm also not sure what does eq (3) mean. Why all the padded actions correspond to state $s_n$. eq(2) defines energy function as pseudolikelihood but then in the Atari experiment, it seems like a different function is used which is associated with reward. Also, if the optimization objective can be anything, why it's called energy?**
>
> N denotes the length of the entire trajectory, and T denotes planning horizon. Rather than directly planning out the entire trajectory, our method recursively plans a segment of next steps. We clarify this difference in Section 3.
> The equation 3 is applied in the test time where we have no knowledge about future states, hence we pad the trajectory with the latest state $s_n$.
> We estimate energy using eq.2 for all environments, but in Atari we add the reward as model context since it yields better performance. We use the terminology of “energy” since the sequence model output is treated as the estimation of energy in the Markov Random Field following previous work [1].
>
> 1. Exposing the Implicit Energy Networks behind Masked Language Models via Metropolis--Hastings. Kartik Goyal, Chris Dyer, Taylor Berg-Kirkpatrick.
>
> **Q3) Minor clarifications for four questions.**
>
> **a) Does $s$ in eq (1) and $s_1$ eq (2) mean? So the "model" is not predicting states?**
>
> No, our model doesn’t predict the states, only predict the actions. To clarify the notation, we have used another symbol $\tau_{ctx}$, which represents the past K steps of states and actions, to replace the $s$ in eq (1) and $s_1$ in eq (2) b.
>
> **b) Following question 1, if that's the case, how can you define constraint based on future states in eq (4)? Are you using the simulator for planning?**
>
> To define constraints -- we use the simulator to predict if an action will cause the agent to enter an invalid state. We use the same constraint definition for both our approach and baselines.
>
> **c) How did you get the value and reward estimation for Atari experiments?**
>
> We get the reward value after each action from the Atari simulator, but we don’t estimate the reward.
>
> **d) What did you get the goal state for BABYAI tasks?**
>
> We set the location $(x,y)$ of the goal object as the goal state in Babyai.
>
> Again, thank you for your comments, and we hope that this addresses your concerns!

---

> > ### Comment · Reviewer_VX64 · 2022-11-18
> > **Response**
> >
> > I would like to thank for authors for the new empirical results and the clarifications.
> > After reading the clarifications and updated version of the paper, I think now I have a better understanding of this work. On the other hand, I also find I might have overestimated the proposed method a bit because:
> >
> > 1. It seems like the planning of LEAP in Online adaptation and non-goal-oriented settings (if not all the settings) relies on access to a simulator. This is a major limitation and should be clearly stated in the paper.
> > 2. The second point is not directly relevant to the authors' response to me. The (updated) standard deviations in Table 3 further undermine the significance of the empirical evaluations. Basically, LEAP performs better than DT only on a single task Breakout and their standard deviation is so high that even 68% CI overlapped with each other. This basically means the only significant evidence that LEAP performs better than DT is on MiniGrid, which is weaker than I expected because the task itself has simple dynamics and that's not the task where DT is developed and tuned.

---

> > > ### Author Response · Authors · 2022-11-18
> > > **Clarifications**
> > >
> > > Thank you for your quick response. I think we may have mis-explained some of responses – please see the following clarifications to your concerns.
> > >
> > > > It seems like the planning of LEAP in Online adaptation and non-goal-oriented settings (if not all the settings) relies on access to a simulator.
> > >
> > > Hi, we realized we incorrectly stated in our response that `we use the simulator to predict if an action will cause the agent to enter an invalid state`. We would actually like to clarify that at **no point in any of our evaluations do we rely on the simulator** in any setting (either online adaptation or non-goal-oriented settings).  Rather, the constraint we pose in the Online Adaptation test is that we  **disable at the current observed state, any immediate actions that cause the agent to enter an invalid state in the simulator**. So if an agent has lava in front of it, it may not execute the forward action at the observed timestep. We do not constrain the agent from making plans of future sequences of actions that enter invalid states (as that would require the use of a simulator), which holds for all other experiments. Rather this setting tests the ability of our policy to generate diverse sets of plans that can adapt to novel circumstances (in this case obstacles) that are suddenly observed by the agent at the current timestep.
> > >
> > > > High standard deviation in Atari environment
> > >
> > > Hi, we would like to clarify that the standard deviations number reported in the paper are the standard deviation values of individual runs in each environment. Thus, it is not correct to directly use the standard deviation values to compare performance of different methods across environments – instead we should compare these reported performance values using standard error (corresponding to standard deviation of estimated mean performance in each environment). We realized this is confusingly presented in the table and we have updated values to report standard error. **Using the standard error, our performance is statistically better than DT by more than 1 standard deviation across 3 separate environments**.

---

> > > > ### Comment · Reviewer_VX64 · 2022-12-02
> > > > **Further response**
> > > >
> > > > Thanks for the further clarification and sorry for the late response.
> > > >
> > > > > We would actually like to clarify that at no point in any of our evaluations do we rely on the simulator ...
> > > >
> > > > OK, that sounds good but I failed to follow your explanation probably because I've lost the context given time passed by. Can you state crisply in which setting (e.g. online adaptation) and in which stage (training/testing) a simulator is needed? To be clear, when we are talking about a simulator, we mean the ground truth dynamics model, not the data, right?
> > > >
> > > > > We realized this is confusingly presented in the table and we have updated values to report standard error.
> > > >
> > > > Yes, I agree using standard error is better, while my previous comments are based on the assumption that you are using standard deviation and I've divided it by $\sqrt{3}$ in my mind. For CI overlaps I didn't mean the CI covers the mean performance of another method but CI overlaps with each other. To improve the significance, an easy way is to increase training seeds, or to use tools like reliable and increase evaluation seeds.

---

> > > > > ### Author Response · Authors · 2022-12-02
> > > > > **Thanks**
> > > > >
> > > > > Thank your for your response -- please see our clarifications below. We're happy to answer any additional questions or clarifications you would like to ask
> > > > >
> > > > > > OK, that sounds good but I failed to follow your explanation probably because I've lost the context given time passed by. Can you state crisply in which setting (e.g. online adaptation) and in which stage (training/testing) a simulator is needed? To be clear, when we are talking about a simulator, we mean the ground truth dynamics model, not the data, right?
> > > > >
> > > > > Yes when talk about the simulator, we mean the ground truth dynamics model. The only setting in which a simulator is used is the online adaptation setting. In this setting -- the simulator is only used, prior to the planning procedure on our learned model, to eliminate any actions from the current state that would result in the agent reaching an invalid next state.  For instance, if the action space of the agent is up, down, left and right but the tile above the agent has lava, then the available actions from the current observed state would just be down, left and right.
> > > > >
> > > > > > Yes, I agree using standard error is better, while my previous comments are based on the assumption that you are using standard deviation and I've divided it by $\sqrt{3}$  in my mind. For CI overlaps I didn't mean the CI covers the mean performance of another method but CI overlaps with each other. To improve the significance, an easy way is to increase training seeds, or to use tools like reliable and increase evaluation seeds.
> > > > >
> > > > > Yes we are in the process of running additional seeds in Atari and will update the final paper with a larger number of random seed runs.

---

### Official Review · Reviewer_QzJV · 2022-10-24

**Confidence:** 3
**Correctness:** 4
**Technical Novelty And Significance:** 3
**Empirical Novelty And Significance:** 3
**Recommendation:** 6

**Clarity, Quality, Novelty And Reproducibility:**

Clarity: This paper is well written and the approach and results are clearly described and illustrated. The results clearly highlight the strengths of the proposed approach, but some further improvements to the ablations and discussions on potential limitations of the proposed method could add further clarity.

Quality & Novelty: The idea of using a language modeling loss to estimate the energy of the data distribution & its use for planning action sequences is interesting and novel. The experiments show promising results, albeit on a limited set of tasks. There are several baseline comparisons and some ablations provided but further ablations can add value.

Reproducibility: The code for the paper is made available on Github and makes it potentially easy to reproduce the results.

**Strength And Weaknesses:**

Strengths:
1. This paper is very well written. The central message of the paper is clearly conveyed and the figures nicely illustrate the different potential applications of the proposed approach.
2. The idea of using a bi-directional language model for energy estimation and using a sampling based approach for computing actions leveraging this energy is interesting and shows good results compared to baselines on a subset of the tested tasks.
3. The approach is extensible and can easily lend itself for compositional objectives which prior methods seem to struggle on.

Weaknesses:
1. A key point mentioned in the motivation of the approach is that "auto-regressive" modeling approaches can struggle to provide good action selections for long-horizon problems due to their uni-directional nature. This was used as a motivation for bi-directional reasoning but it is not clear if this makes a difference as there are no baselines that ablate this selection (I presume a different planning method would also be needed for this ablation).
2. The key novelty (in my opinion) of the presented approach is the choice of masked language modeling style losses for modeling the expert data distribution, and subsequent use of this likelihood function for planning. Which of these two matter more? Would any generic planning approach (e.g. random shooting vs Gibbs sampling vs cross-entropy style planning) work with the proposed energy function? And vice-versa would other ways of approximating the energy work with the proposed planning approach? This is unclear from the current results, and an ablation for this would be very illuminating.
3. The results on Atari are quite weak; in fact, CQL performs better than the proposed approach on 3/4 tasks, and only on a single task the approach gets a high final score which makes the overall score be higher than CQL. Additionally, the results in Fig 7 indicate that while there is some correlation between the achieved rewards and the estimated energy there are a significant number of outliers. Why is this the case?
4. There are three specific extensions of the proposed approach mentioned: online adaptation, novel environment generalization and task compositionality. It is not clear if these are unique to the presented approach. Specifically, the proposed online adaptation schema can work for any sampling based planning approach if rejection sampling is used, and the task compositionality only works if the energy functions can be linearly combined (and even then there can be significant local optimas from the planning perspective). These should be highlighted in the presentation.
5. A potential limitation of the proposed approach is its applicability to continuous action spaces; several, if not all of the presented baselines can directly be applied to continuous action spaces whereas the proposed approach is limited to discrete tokens. This can potentially be relaxed by binning/discretizing continuous actions but nonetheless it would be useful to discuss this in the main text.
6. It would useful to understand the computational/time complexity of the proposed planning approach as it requires several sequential iterations with a potentially large model for successful planning.

**Summary Of The Paper:**

This paper presents an approach to leveraging a language modeling objective to train an energy function which can be used together with a sampling procedure to generate trajectories for discrete-control tasks. The approach uses a masked language modeling loss to train a bi-directional language model to fit expert demonstrations; the likelihood of this model is used to choose actions (given a context of states and actions) via an iterative Gibbs sampling procedure. The method is tested on a subset of BabyAI and Atari tasks and compared to several baselines; performance is significantly better than baselines on the BabyAI tasks and about the same as baselines on Atari tasks. Extensions of the approach to enable solutions to compositional problems are also propose and show good results on the tested tasks.

**Summary Of The Review:**

Overall, this paper presents an interesting new approach to utilizing a language modeling loss to estimate an energy for planning action sequences and is tested on a small set of tasks in two different domains. The results are promising and show improvements compared to the presented baselines. I would suggest a borderline accept.

---

> ### Author Response · Authors · 2022-11-15
> **Reviewer QzJV Response**
>
> We thank Reviewr QzJV for your careful review. We have addressed each of your concerns in the response below and have also updated the paper accordingly.
>
> **Q1) Unidirectional baselines**
> In unidirectional setting, earlier action predictions are not dependent on future action predictions. Since there is no dependence of future actions on early actions, generating trajectories in this setting corresponds to autoregressive action prediction. This corresponds to the decision transformer baseline.
>
> **Q2) Ablations of Energy Function / Planning Approach.**
>
> Thank you for the suggestion! It’s interesting to try these combinations, we have added the 6 different combinations of planning methods (random shooting, cross-entropy style planning and Gibbs sampling) and energy models (masked language models and sequence model classifier). We describe this performance in Appendix C. We observe that the Gibbs sampling gives the best outcome (with CEM a close second) with the MLM model and that defining an energy value using a sequence model classifier doesn't work well in all settings.
>
> | Energy model | Random-shooting | CEM | Gibbs |
> | ------------- | ------------- | ------------- | ------------- |
> | MLM  | 23.3% | 57.5% | **62.5%** |
> | Classifier  | 25.0% | 12.5% | 15.5% |
>
> **Q3) Atari performance and outliers.**
>
> While our we agree that our approach isn’t as good as CQL across all tasks – our approach represents the first work exploring planning in sequence modeling using energy minimization, and still outperforms the strong sequence modeling baseline of the decision transformer (and our approach is online outperformed by CQL on one environment).
> In terms of outliers with respect to energy/reward correlation, our underlying energy function is trained using maximum likelihood, and thus tends to assign low energy to frequently occurring trajectories in the dataset. Thus, since our model is not directly trained to assign low energy to high reward trajectories the correlation between rewards and energy values is a bit noisy.
>
> **Q4) There are three specific extensions of the proposed approach mentioned: online adaptation, novel environment generalization and task compositionality. It is not clear if these are unique to the presented approach. Specifically, the proposed online adaptation schema can work for any sampling based planning approach if rejection sampling is used, and the task compositionality only works if the energy functions can be linearly combined (and even then there can be significant local optimas from the planning perspective). These should be highlighted in the presentation.**
>
> We have clarified in the main text of the paper that our approach “naturally enables these three properties” as opposed to “uniquely enables these three properties”. While rejection sampling may be used to adapt sampling based approaches, such a procedure would require a large number of samples to find a valid constrained samples – in contrast constraints are directly integrated in sample generation in our approach. We have clarified this in the method section. We have further clarified that task compositionality works when adding energy functions when each composed subgoal is independent with respect to each other (which we believe occurs in a variety of separate tasks).
>
> **Q5) Applicability to continuous action spaces**
>
> Indeed, a current limitation of our approach is that it has to be applied to discrete action spaces– we’ve added in the conclusion that this is a promising direction of future work. In principle, it would not be difficult to extend our continuous action spaces – for instance we may use a discrete binning system used in [1].
>
> 1. Reinforcement Learning as One Big Sequence Modeling Problem. Michael Janner, Qiyang Li, Sergey Levine.
>
> **Q6) Computational/time complexity of LEAP.**
>
> We have clarified in section 3.2 and algorithm 1 that the computational time consumption increases with more iteration numbers. To be more specific, the increment is linear, from $O(n^2)$ of the single forward pass of a transformer to $O(Nn^2)$, where $n$ stands for input sequence length and $N$ for iteration number. While the overhead exists, the total time consumption could still be limited on a modern GPU. Furthermore, we have shown in Figure 6 that the model performance improves with more iteration number. Hence in application, we can easily control the tradeoff between time consumption and performance by altering the iteration number.
>
> Again, thank you for your comments, and we hope that this addresses your concerns!

---

> > ### Comment · Reviewer_QzJV · 2022-11-19
> > **Response**
> >
> > Thanks to the authors for addressing my comments in the rebuttal, particularly the ablation on energy function & planning approaches. There are still some questions re the performance as highlighted by other reviewers so I'll keep my current score as is but am overall positive on the paper.

---

### Official Review · Reviewer_MKux · 2022-10-24

**Confidence:** 3
**Correctness:** 2
**Technical Novelty And Significance:** 2
**Empirical Novelty And Significance:** 2
**Recommendation:** 8

**Clarity, Quality, Novelty And Reproducibility:**

As discussed above there are quite a few details that are unclear which makes it hard to completely evaluate the method. In terms of novelty, there are quite a few works that have looked at using a transformer based model for predicting action sequences. Some of which are mentioned in the paper, a few relevant ones like [1] are overlooked (I would definitely urge the authors to look at [1] as it does touch up on the issue of handling stochastic setting). However to the best of my knowledge, their specific approach to sampling sequence is novel.

[1] Paster, Keiran, Sheila McIlraith, and Jimmy Ba. "You Can't Count on Luck: Why Decision Transformers Fail in Stochastic Environments." arXiv preprint arXiv:2205.15967 (2022).

**Strength And Weaknesses:**

One of the main strengths of the approach is the fact that it presents a straightforward application of existing learning methods.

In terms of weaknesses, one of the central problems is the clarity of exposition. In fact, the current description of the method presented in the paper is not sufficient to clearly evaluate the properties of the method. First off, the method is referred to as planning. In the common usage of the term, planning usually involves reasoning with some form of task model. It isn’t completely clear if there is a task model being maintained in this case. Is it referred to as planning because you are using an energy function or is the probabilistic model meant to stand for the world model?

Also assuming access to a set of near optimal training data brings the setting closer to a behavior cloning set up than traditional RL set up, where the agent has to learn through interaction with the environment. The fact that for the experiments, you had to initially run an RL agent (at least in atari) kind of speaks against the usefulness of these methods on their own. Also the energy function itself makes no reference to total reward associated with each trajectory. If the original trajectory set contained trajectories of varying quality how does the model differentiate them during the testing time? Would it just have to assume the better action showed up with higher frequency?

It is also unclear to me if the method was only designed for deterministic settings? For, stochastic settings solutions can’t take the form of a sequence of actions, rather it needs to take the form of policies. Which means after each action, it needs to check what state it leads to and decide the next action based on this information. While in theory the model can support such execution, it is unclear if one really benefits from the use of such a sequential model in stochastic settings. After all the sequence predicted by the model, even if it is the highest likelihood one, may end up not being followed.  Instead at each step, one needs to consider the expected value associated with following a given action. However as discussed above, the model doesn’t really seem capable of associating any kind of value with a specific action.

Finally on a smaller note, while the learning methods itself uses tools from NLP that were used to learn language models, they are not used to learn language models in this context. So I might advise the authors from referring to the method as planning using language models and maybe just refer to the fact that you are using sequential models.

In terms of experiments, as discussed the generation of action trajectories are a bit unrealistic, but apart from that the experiments presented seem like a reasonable starting point. One point I would like to know is whether the model presented here can perform better than the traces that were present originally. In other words, if it was presented a state and context that correspond to an example trace part of the initial set, could it have come up with a better course of action by leveraging other traces it saw.

Post Rebuttal Comment: After the new set of experiments and the detailed discussion with the authors. I am happy to recommend the paper for acceptance.


**Summary Of The Paper:**

The paper looks at the use of masked-language models trained on a given set of trajectories to generate action sequences that minimize a specified energy function. If the energy function captures the objectives of a sequential decision-making problem, then the procedure could be utilized to generate the course of action to be performed. The proposed system is then tested in various domains including, BabyAI and ATARI and shows a number of desirable properties such as online adaptation,  novel environment generalization and task compositionality.

**Summary Of The Review:**

While I think the paper presents an interesting direction and formulation, I don’t believe the paper in its current form is ready to be published yet.

---

> ### Author Response · Authors · 2022-11-15
> **Reviewer MKux Response**
>
> We thank Reviewr MKux for your careful review. We have addressed each of your concerns in the response below and have also updated the paper accordingly.
>
> **Q1) Is the planning model probabilistic**
>
> We have clarified in section 3 that the planning procedure refers to the iterative energy minimization procedure through which we obtain an optimal minimal trajectory – our energy function is not meant to be a probabilistic model which stands for the world model.
>
> **Q2) The problem setting is different from traditional RL setup**
>
> It is true that the application of our non-reward conditioned agent would require demonstration data obtained by a RL agent. While this does depart from the typical reinforcement learning setting, we demonstrate in the paper that our learned energy function can then  enable successful planning in both harder environments and environments with novel constraints. In our paper, we refer to our method as planning and not reinforcement learning. Furthermore, as discussed in Q3, our approach can be applied in the more traditional RL setup and be conditioned on total reward, with the reward conditioning used to refine behaviors in the dataset.
>
> **Q3) The energy function itself makes no reference to total reward associated with each trajectory, hence cannot distinguish between the quality of demonstrations.**
>
> We would like to clarify that our approach can easily be conditioned on total reward, by simply concatenating the reward as input in the sequence model. In fact, as discussed in original paper Appendix A.4, our model takes the reward as input following DT in the Atari environments, so that the model is capable of recognizing the quality of training trajectories – and potentially improving them. We further validate this by adding an experiment to show the performance degradation WITHOUT the reward input, which is demonstrated in Appendix A.4, indicating that approach is refining and not just imitating the information in a dataset.
>
> | Env | w/o return | w return |
> | ------------- | ------------- | ------------- |
> | Breakout | 182.0 | 378.9 |
> | Qbert | 41.0 | 19.6 |
> | Pong | 100.7 | 108.9 |
> | Seaquest | 0.5 | 1.3 |
>
> **Q4) LEAP for stochastic settings?**
>
> We believe our approach can be applied to stochastic settings. Our training method trains the sequence model to assign low energies to trajectories with higher likelihood of success, as long as they have higher appearing frequency in the dataset. Therefore, in the stochastic environments, it constructs a sequence of actions that has the highest likelihood to reach the goal. When executing this plan in a stochastic environment, we may then choose to replan our sequence of actions after each actual action in the environment (to deal with stochasticity of the next state given an action). Note that this sequence of actions will be optimal in the stochastic environment, as we always choose the action that has the maximum likelihood of reaching the final state.
> Note that this behavior is actually better than using a simple policy to predict the next action in the environment as it simply assigns probability distribution to the immediate next step without the awareness of future step adjustments facing stochastic factors. To verify this conclusion,  we have designed a stochastic experiment in the BabyAI, where the actions have a random chance to fail (details in Appendix B). In this setting, we find that our approach, where we plan a sequence of actions in a stochastic environment obtains a success rate of 57.5% while a policy (Decision Transformer) obtains a much poorer performance of 30.8%. More extensive results and discussions are found in the Appendix B. We have also added a citation to the relevant paper.
>
> **Q5) planning using language models**
>
> Yes, this is a great point. We have revised the title of our paper to be “Planning with Sequence Models through Energy mInimization”.  We have further replaced instances of the word language in the paper with sequence.
>
> **Q6) Can the model perform better than the traces that were present originally in dataset?**
>
> Yes, we believe our approach can perform better than the traces originally present in the dataset. As mentioned earlier in our response, in the Atari setting, by conditioning on the reward of each trajectory – we are able to construct better behaviors (Table 10 in Appendix A.4). Furthermore, note that both in Table 1 and 3, our approach substantially outperforms a behavior cloning (BC) method, which directly learns by copying the information in traces from the dataset.
>
> Again, thank you for your comments, and we hope that this addresses your concerns!

---

> > ### Comment · Reviewer_MKux · 2022-11-30
> > **Re: Reviewer MKux Response**
> >
> > I apologize for the delayed response. I appreciate the authors efforts in adding a new domain, it definitely improves the significance of the results presented. However, I did have some questions with respect to some of the comments the authors made
> >
> > >with higher likelihood of success, as long as they have higher appearing frequency in the dataset.
> >
> > Are the requirements here at a trajectory level? or at the level of individual transitions (i.e., frequency of transitions reflect their likelihood)?
> > If it's the former, it feeds into one of my main concerns with the paper that the method already requires access extremely informative demonstrations.
> >
> > > Note that this sequence of actions will be optimal in the stochastic environment, as we always choose the action that has the maximum likelihood of reaching the final state
> >
> > I am unsure, why this should be the case. You might have actions which has extremely high probability of reaching the goal state, with a small probability leading to an absorbing non-goal state. Such actions might in fact be part of the most likely trajectory to the goal, but from would be the wrong action to pick from an expected value point of view. I am not sure how your method would be able to handle those cases.
> >
> > > Note that this behavior is actually better than using a simple policy to predict the next action in the environment as it simply assigns probability distribution to the immediate next step without the awareness of future step adjustments facing stochastic factors.
> >
> > I would really appreciate a bit more clarity on this claim. After all, one can prove that a deterministic non-stationary policy can capture optimal behavior within an infinite-horizon discounted MDP. Are you saying this is not true?

---

> > > ### Author Response · Authors · 2022-11-30
> > > **Clarifications**
> > >
> > > Thank you for your comments and feedback. Please see our responses below:
> > >
> > > > Are the requirements here at a trajectory level? or at the level of individual transitions (i.e., frequency of transitions reflect their likelihood)? If it's the former, it feeds into one of my main concerns with the paper that the method already requires access extremely informative demonstrations.
> > >
> > > The requirements would be only that local patterns of trajectories be seen frequently. The masked language modeling objective focuses on reconstructing masked portions of trajectories given surrounding context. Since such trajectory reconstruction depends primarily on only nearby timesteps of a given masked timestep, the masked language model primarily captures/repeats local patterns in trajectories.
> > >
> > > > I am unsure, why this should be the case. You might have actions which has extremely high probability of reaching the goal state, with a small probability leading to an absorbing non-goal state. Such actions might in fact be part of the most likely trajectory to the goal, but from would be the wrong action to pick from an expected value point of view. I am not sure how your method would be able to handle those cases.
> > >
> > > Yes, let us clarify this statement -- assuming a perfectly trained instance of model, our masked language model captures perfectly the marginal action distribution $\log p(a_i|s, g, a_{-i})$ given observed state $s$ and goal state $g$. Planning by minimizing the energy function defined by this masked language model corresponds then to finding the set of actions so that the likelihood $p(s, g, a_1, a_2, \ldots, a_T)$ is maximized -- so this corresponds to finding the sequence of actions that is most likely to reach the goal state $g$ given a particular start state $s$ -- assuming a fixed planning horizon of length T.  This means our method would pick the most likely trajectory to the goal -- assuming we have exactly $T$ steps of actions to reach the goal. It would not handle the case you mentioned -- as this would involve more than $T$ action steps in the environment. However, we can still resolve this issue by increasing the planning horizon length.
> > >
> > > > I would really appreciate a bit more clarity on this claim. After all, one can prove that a deterministic non-stationary policy can capture optimal behavior within an infinite-horizon discounted MDP. Are you saying this is not true?
> > >
> > > Yes, we should clarify this claim -- you are correct that in principle a deterministic non-stationary policy can also capture the optimal behavior in order to act in a stochastic environment. What we meant to say is that it may be easier for our model to learn to predict accurate behavior in stochastic dynamics. Our reasoning is that in the stochastic setting, our model can learn to plan / iteratively refine a sequence of actions which will lead to a final goal state with high likelihood. In contrast, a policy must, in one feedforward pass, predict an action that enables all future actions to accurately reach the goal, accounting for all stochasticity of dynamics at state in the future. Such a computation seems difficult to amortize using a learned model.

---

> > > > ### Comment · Reviewer_MKux · 2022-12-01
> > > > **Re: Re: Clarification**
> > > >
> > > > Thank you for the quick response. Unfortunately, your response brings up even more questions.
> > > >
> > > > > The requirements would be only that local patterns of trajectories be seen frequently
> > > >
> > > > So by local patterns, I am assuming you require more than transition level frequencies. If that is the case, then Model-based methods would have an advantage in this case, as they would only need to have dataset that reflect individual probabilities and planner would be able to figure out the best path to the goal. I know you compared against a model-based method in the context of deterministic case, but is there a reason to believe it would hold in the context of stochastic problem? I can at least make a case here that model-based methods may be more sample efficient.
> > > >
> > > > >However, we can still resolve this issue by increasing the planning horizon length.
> > > >
> > > > I am a bit confused by this claim. You can have the problem occur in a single step plan. Assume there is one action which with high probability takes you to the goal, but with a small probability take you to a sink state. Wouldn't your method select this action? However, if you were to consider a method that can reason about expected values of the action, it wouldn't take this action.
> > > >
> > > > >Our reasoning is that in the stochastic setting, our model can learn to plan / iteratively refine a sequence of actions which will lead to a final goal state with high likelihood
> > > >
> > > > Is this based on any previous results? Are there experiments in the paper that reflect this fact?
> > > >
> > > > On a smaller note, I saw that in your response to other reviewers you had mentioned that you don't use a simulator. Then by what convention does your method correspond to a planning technique? Are there any fields that refers to trajectory generation through energy minimization without a world model as planning?

---

> > > > > ### Author Response · Authors · 2022-12-02
> > > > > **Additional Clarifications**
> > > > >
> > > > > Thanks you for quick response -- please see our clarifications below. As always, we are happy to chat more and clarify any other questions you have.
> > > > >
> > > > > > So by local patterns, I am assuming you require more than transition level frequencies. If that is the case, then Model-based methods would have an advantage in this case, as they would only need to have dataset that reflect individual probabilities and planner would be able to figure out the best path to the goal. I know you compared against a model-based method in the context of deterministic case, but is there a reason to believe it would hold in the context of stochastic problem? I can at least make a case here that model-based methods may be more sample efficient.
> > > > >
> > > > > While we also agree that it is true that a model-based method would have an advantage in accurately capturing environmental dynamics (as they only need individual probabilities), whether this would lead to more effective planning is not completely clear. When planning with a model-based method, individual state are autoregressively rolled out and thus longer plans are more likely to exploit adversarial modes in the dynamics model (as for example discussed in [2]). In contrast, planning by learning a model across local sequences of actions (as also discussed in [2]) would be more be less prone to adversarial modes of the dynamics model.
> > > > >
> > > > > Thus while we are happy to run an additional comparison with model-based methods in the final version of the paper -- we are unsure if there would be an underlying reason why the trend of our approach outperforming model based methods would not hold in the stochastic setting.
> > > > >
> > > > > > I am a bit confused by this claim. You can have the problem occur in a single step plan. Assume there is one action which with high probability takes you to the goal, but with a small probability take you to a sink state. Wouldn't your method select this action? However, if you were to consider a method that can reason about expected values of the action, it wouldn't take this action.
> > > > >
> > > > > Sure -- let's consider an environment in which action A has high probability to reach the goal, but a small probability to reach a sink state. In this same environment, let action B, when executed twice reach the goal with 100% probability.
> > > > >
> > > > > In the setting, if our planning horizon is 1 -- the highest likelihood action to reach the goal would indeed be to pick action A. However, now consider if our planning horizon is length 2 -- the highest likelihood sequence of actions to reach the goal would now be pick Action B. In general, by planning for a long number of timesteps -- we can circumvent the need of using the expected value of actions.
> > > > >
> > > > > > Is this based on any previous results? Are there experiments in the paper that reflect this fact?
> > > > >
> > > > > Yes, in main paper Figure 6, we illustrate that our method correctly captures the energy landscape across different trajectories and how it may successfully finds a minimal energy (high-likelihood) trajectories by running larger number of planning iterations. While both experiments are done in a deterministic setting -- our training objective / planning formulation is adapted from the NLP literature [1] which is a highly stochastic setting, so it seems likely that a similar observation would hold in the stochastic setting.
> > > > >
> > > > > > On a smaller note, I saw that in your response to other reviewers you had mentioned that you don't use a simulator. Then by what convention does your method correspond to a planning technique? Are there any fields that refers to trajectory generation through energy minimization without a world model as planning?
> > > > >
> > > > > In robotics -- trajectory optimization (which corresponds to energy minimization across a set of cost functions) is often referred to as planning (for example see https://rll.berkeley.edu/trajopt/doc/sphinx_build/html/).
> > > > >
> > > > > [1] Kartik Goyal et. al. Exposing the Implicit Energy Networks behind Masked Language Models via Metropolis--Hastings
> > > > >
> > > > > [2] Janner et. al. Planning with Diffusion for Flexible Behavioral Synthesis

---

> > > > > > ### Comment · Reviewer_MKux · 2022-12-02
> > > > > > **Re: Re: Re: Clarification**
> > > > > >
> > > > > > >However, now consider if our planning horizon is length 2 -- the highest likelihood sequence of actions to reach the goal would now be pick Action B.
> > > > > >
> > > > > > There is still a problems here, right? I am assuming you would be doing an iterative increase of the horizon length. If this is the case, you would pick the A and go with it, with potentially disastrous results.
> > > > > >
> > > > > > >In robotics -- trajectory optimization (which corresponds to energy minimization across a set of cost functions) is often referred to as planning (for example see https://rll.berkeley.edu/trajopt/doc/sphinx_build/html/).
> > > > > >
> > > > > > Wouldn't trajectory optimization in robotics be using a world/robot model? Otherwise, how would you account for potential collisions and safe joint angles?

---

> > > > > > > ### Author Response · Authors · 2022-12-02
> > > > > > > **More Clarifications**
> > > > > > >
> > > > > > > Thanks again for your quick response. Please see our clarification below.
> > > > > > >
> > > > > > > > There is still a problems here, right? I am assuming you would be doing an iterative increase of the horizon length. If this is the case, you would pick the A and go with it, with potentially disastrous results.
> > > > > > >
> > > > > > > The above example in which we iteratively increase the horizon length is a didactic example illustrating differences in behavior when the planning length is changed. In practice, at test time, you could pre-specify a planning length roughly equal to the task completion time (which you could obtain from training data) which you would then use to generate all plans.
> > > > > > >
> > > > > > > > Wouldn't trajectory optimization in robotics be using a world/robot model? Otherwise, how would you account for potential collisions and safe joint angles?
> > > > > > >
> > > > > > > In trajectory optimization -- you infer a set of actions for a trajectory which minimize a set of defined cost functions across actions. Certain cost functions operate directly on inferred actions (such as enforcing smoothness across individual actions). Other cost functions enforce feasibility of actions, such as collision avoidance and joint limits, by using a simulator to infer resultant states after executing actions and then enforcing that each resultant state is a valid. In our setting, we use an analogous idea of cost function optimization across actions, but all our cost functions are learned.  We use a learned model to enforce the feasibility of actions, without the explicit use of the simulator.

---

> > > > > > > > ### Comment · Reviewer_MKux · 2022-12-03
> > > > > > > > **Re: More Clarifications**
> > > > > > > >
> > > > > > > > First off, I really appreciate the authors willingness to engage in a dialogue and this really reflects the strengths of openreview as a reviewing platform. However, if I could ask one more clarifying question.
> > > > > > > >
> > > > > > > > > roughly equal to the task completion time (which you could obtain from training data)
> > > > > > > >
> > > > > > > > For the example we were discussing, the training data would include the one step plan with higher frequency. Thus if you were to estimate completion time from the training data, wouldn't you estimate it to be one step?

---

> > > > > > > > > ### Author Response · Authors · 2022-12-03
> > > > > > > > > **Additional Clarifications**
> > > > > > > > >
> > > > > > > > > Thank you for taking time to engage with us during the discussion period. Please see our response below:
> > > > > > > > >
> > > > > > > > > > For the example we were discussing, the training data would include the one step plan with higher frequency. Thus if you were to estimate completion time from the training data, wouldn't you estimate it to be one step?
> > > > > > > > >
> > > > > > > > > One option to estimate completion time from training data would be to choose the maximum episode length in the training data -- which in this setting would be two. This heuristic would be useful in a large number of RL environments as they typically have a fixed maximum length of episodes.

---

> > > > > > > > > > ### Comment · Reviewer_MKux · 2022-12-03
> > > > > > > > > > **Re: Additional Clarifications**
> > > > > > > > > >
> > > > > > > > > > But maximum horizon length would overlook actual optimal paths, unless you use noop actions (i.e., dummy actions that are used to pad the length of the sequence). But if you use noop actions, your system is now back to selecting the highly likely but dangerous path (because you can do the dangerous action and pad the length using noop actions, which will be the most likely plan). In fact, this problem corresponds to a well-studied problem within probabilistic planning community, that is used as an argument against the use of action sequences as solutions to probabilistic planning. [1] is a good reference point to understand the underlying issues. I still don't see how your system would be able to handle this problem, but would be happy to be proven wrong.
> > > > > > > > > >
> > > > > > > > > > [1] Little, Iain, and Sylvie Thiebaux. "Probabilistic planning vs. replanning." ICAPS Workshop on IPC: Past, Present and Future. 2007.

---

> > > > > > > > > > > ### Author Response · Authors · 2022-12-03
> > > > > > > > > > > **Additional Clarifications**
> > > > > > > > > > >
> > > > > > > > > > > Hi, we think it is actually possible to construct such a probabilistic planning procedure. The main intuition is that adding a  noop action never increases the likelihood of the underlying plan reaching a particular goal. Please read the description below -- also happy to discuss more.
> > > > > > > > > > >
> > > > > > > > > > > Consider the simulated environment we were discussing earlier, where we have action A which directly gets you to the goal with probability 95% and a sink state with probability 5%, action B which gets to the goal state with probability 100% after 2 steps, and now a newly defined noop action C which simply keeps you at a particular state you are at with 100% probability. Let's now let's look at the probability of success of policies length 2. The probability of reaching the goal using action A would be Pr(AC) = Pr(A) * Pr(C) = 0.95 * 1.0 = 0.95 while the probability of reaching the goal using action B would Pr(BB) = 1.0 and thus the maximum likelihood plan is still executing action B.

---

> > > > > > > > > > > > ### Comment · Reviewer_MKux · 2022-12-03
> > > > > > > > > > > > **Re: Additional Clarifications**
> > > > > > > > > > > >
> > > > > > > > > > > > You seem to have misunderstood my question. In the scenario I am talking about Pr(A) remains the most likely trajectory period. But using B is a better option, because they don't ever lead to a trap states. So there is always a path with non-zero probability to reach the goal. But none of those trajectories, will have probability higher than Pr(A).

---

> > > > > > > > > > > > > ### Author Response · Authors · 2022-12-03
> > > > > > > > > > > > > **Clarifications**
> > > > > > > > > > > > >
> > > > > > > > > > > > > Hi, why would Pr(A) be the most likely trajectory period? Assuming we have a perfectly learned model -- the model would learn the marginal distributions of goal states given each action. So since A sometimes goes to the goal states and sometimes to a sink state it would have lower likelihood than Pr(B) which always goes to the goal state.

---

> > > > > > > > > > > > > > ### Comment · Reviewer_MKux · 2022-12-03
> > > > > > > > > > > > > > **Re: Clarifications**
> > > > > > > > > > > > > >
> > > > > > > > > > > > > > But you are looking at likelihood of trajectories. Consider a case where there are only three possible trajectories to goal
> > > > > > > > > > > > > > A - results in goal with 0.95
> > > > > > > > > > > > > > B - Two stochastic outcomes with likelihood 0.5 and 0.5
> > > > > > > > > > > > > > If it results in outcome1 you can use action C to get to goal with probability 1 and if it's outcome 2 you can use action D to get to goal with probability 1.
> > > > > > > > > > > > > >
> > > > > > > > > > > > > > So the probability of the goal reaching trajectories are
> > > > > > > > > > > > > >
> > > > > > > > > > > > > > Pr(A) - 0.95
> > > > > > > > > > > > > > Pr(B,C) - 0.5
> > > > > > > > > > > > > > Pr(B,D) - 0.5

---

> > > > > > > > > > > > > > > ### Comment · Reviewer_MKux · 2022-12-03
> > > > > > > > > > > > > > > **Re: Re: Clarifications**
> > > > > > > > > > > > > > >
> > > > > > > > > > > > > > > From the initial read to the rebuttal, I kind of forgot that you are in fact marginalizing across the trajectories. That would in fact address my major concerns. I will update my scores to reflect that fact, thank you for patiently walking me through my concerns.

---

> ### Author Response · Authors · 2022-11-28
> **Looking Forward to Your Reply**
>
> Dear Reviewer MKux,
>
> Thank you for taking time to read and review our paper. We have followed your suggestions and have ran additional experiments on stochastic settings and illustrating behavioral refinement. We have also clarified and modified the text following your suggestions.
>
> As the discussion period is nearing the end, we would appreciate if you could kindly check our response. Please don’t hesitate to ask us if you have any more questions.
>
> Thanks,
> Authors

---

### Official Review · Reviewer_gPMg · 2022-10-25

**Confidence:** 4
**Correctness:** 3
**Technical Novelty And Significance:** 3
**Empirical Novelty And Significance:** 2
**Recommendation:** 6

**Clarity, Quality, Novelty And Reproducibility:**

I found the paper well written and easy to follow. I think the planning with bidirectional language models for offline RL is interesting. While most implementation details are given, some pieces are unclear to reproduce results such as transformer attention mechanism.

**Strength And Weaknesses:**

**Strengths** The paper is written well and easy to follow through. I found the iterative planning with bidirectional LMs interesting and broadly applicable to other settings including multi-agent learning (not mentioned in the paper).

**Weaknesses**

1. There are no model-based or autoregressive planning baselines. For example, *PlaTe* and *MuZero Unplugged* are two baselines that need to be cited and compared against.

2. Baselines for the Lava experiment are used as is while a large energy is added for those states in LEAP; which a bit unfair in my opinion. Could you add baselines where you just mask an action for DT/IQL if the action is leading to a lava?

3. At the beginning of section 3, in both argmax and argmin, you don't have $s_{2:T}$. Was this intentional or just a mistake in writing? If it was intentional, could you explain why you only condition on the first state in your formulation? You use only the first action in Atari, so I am assuming this doesn't apply to it.

4. Please add confidence intervals to Table-3. DT results have wide intervals and it is difficult to compare LEAP to DT without error bounds.

5. Can you discuss if LEAP would work for non-deterministic dynamics?

6. In Appendix A.2. you mention causal attention while in main text you mention bidirectional attention with future. Please clarify if you use bidirectional or causal attention.

7. You mention that $\tau^*$ is derived by minimizing $E_\theta(\tau)$ but this is not true as your model doesn't predict states. Please clarify.

8. "baseslines" --> "baselines",
"adaptaiton" --> "adaptation"


PlaTe: Visually-Grounded Planning with Transformers in Procedural Tasks. Jiankai Sun, De-An Huang, Bo Lu, Yun-Hui Liu, Bolei Zhou, Animesh Garg.

Online and Offline Reinforcement Learning by Planning with a Learned Model. Julian Schrittwieser, Thomas Hubert, Amol Mandhane, Mohammadamin Barekatain, Ioannis Antonoglou, David Silver.

**Summary Of The Paper:**

This paper studies planning with language models (LM) using iterative energy minimization. The authors utilize a bidirectional LM with the masked language model (MLM) objective. The model is trained by optimizing for MLM using expert trajectories. For inference, they iteratively mask actions in a trajectory, including future actions, and use Gibbs sampling to iteratively predict and refine actions. The authors use a decision transformer (DT) for the encoding of a trajectory and explain benefits of this iterative planning approach. Empirical results on BabyAI and Atari games show that LEAP outperforms previous offline RL methods on BabyAI and it is in par with DT on Atari.

**Summary Of The Review:**

I think bidirectional LM based planning with Gibbs sampling is interesting. I found some baselines missing, especially relevant model-based and planning baselines.

---

> ### Author Response · Authors · 2022-11-15
> **Reviewer gPMg Response**
>
> Thank you for your detailed comments. We have added two model-based planning benchmarks, tested LEAP in stochastic environments, and added clarifications about the methods and experiments settings.
>
> **Q1) No model-based or autoregressive planning baselines.**
>
> We agree that the suggested experiments will add to the paper significantly, and we tested two model-based baselines PlaTe [1] and MOPO[2] whose results are collected in Table1. Unfortunately the experimental code for MuZero Unplugged is not released. Instead, we add MOPO[2] as another baseline. We have cited and discussed PlaTe, MuZero and MOPO in the related works section of our paper.
> | Task | Env | LEAP | PlaTe | MOPO |
> | ------------- | ------------- | ------------- | ------------- | ------------- |
> | TP | GoToLocalS7N5 | **89.5%** | 42.5% | 87.0% |
> | TP | GoToLocalS8N7 | **89.0%** | 45.0% | 81.5% |
> | TP | GoToObjMazeS4 | **65.0%** | 35.5% | 60.0% |
> | TP | GoToObjMazeS7 | **45.5%** | 30.5% | 30.5% |
> | IC | PickUpLoc | **65.0%** | 7.5% | 43.5% |
> | IC | GoToSeqS5R2 | **38.0%** | 25.0% | 30.0% |
> | IC | GoToObjMazeS4R2Close | **55.5%** | 32.5% | 36.0% |
>
> 1. PlaTe: Visually-Grounded Planning with Transformers in Procedural Tasks. Jiankai Sun, De-An Huang, Bo Lu, Yun-Hui Liu, Bolei Zhou, Animesh Garg.
> 2. Mopo: Model-based offline policy optimization. Yu T, Thomas G, Yu L, Ermon S, Zou JY, Levine S, Finn C, Ma T.
>
> **Q2) LEAP for stochastic dynamics?**
>
> We believe our approach can be applied to stochastic settings. Our training method trains the sequence model to assign low energies to trajectories with higher likelihood of success, as long as they have higher appearing frequency in the dataset. In contrast, when  applying the Decision Transformer to stochastic environments – it simply assigns probability distribution to the immediate next step without the awareness of future step adjustments facing stochastic factors. To verify this, we design a stochastic experiment in the BabyAI and compare our approach against Decision Transformer(DT). The results and discussions are collected in the Appendix B.
> | Env | LEAP | DT |
> | ------------- | ------------- | ------------- |
> | GoToObjMazeS4 | 57.5% | 30.8% |
> | GoToObjMazeS7 | 33.3% | 28.3% |
>
> **Q3) Clarification: Model is not predicting states, hence model is not minimizing E(tau)**
>
> We have clarified in section 3 that $\tau = (\tau_{ctx}, a_{1:T})$ and the optimization objective is to find $a_{1:T}$  to achieve the lowest energy.
>
> **Q4) Questions about method details.**
>
> **a) Please clarify if you use bidirectional or causal attention.**
>
>  For all our experiments we use bidirectional except for Atari where we found casual attention to perform better.
>
> **b) Don't have S_2:T in in both argmax and argmin.**
>
> This is intentional. Because the task is to predict the future actions/states based on the current and/or past action and state. In the formula we use the S_1 to represent the current state, while S_2:T to represent future states. The function is conditioned on the first state since we don’t have access to future states at the stage of planning.
>
> **c) Add confidence intervals to Table-3.**
>
> Yes, we have added the variance across 3 seeds for LEAP and DT in Table-3.
>
> **d) Baselines in lava experiments**
>
> We would like to clarify that for all baselines including DT/IQL, we have already removed the immediate action that will lead to a lava. So the comparison is fair. We have clarified this in section 6.1.
>
> Again, thank you for your comments, and we hope that this addresses your concerns!

---

> > ### Comment · Reviewer_gPMg · 2022-11-28
> > **Thank you for the additional experiments and clarification.**
> >
> > R4-c) Thanks for the confidence intervals. The intervals for DT and LEAP overlap for the first three environments which makes me wonder if the results are really significant. Given that Atari is the only partially observable testbed that you used, the results are not very convincing that LEAP will generalize to more realistic settings. Was there a particular reason why you used fully observable setting in BabyAI?
> >
> > R1) Thanks for the additional baselines. Could you please discuss in more detail why your method performs favorably compared to both model-based baselines? Why do these algorithms fail when LEAP is not failing? Could you also clarify if you are also using whole environment state for both of these methods?
> >
> > R4-b) It would help if you use the index consistently where (1) denotes the beginning of an episode, (2) next state, etc. or alternatively add (t) for current state like (t+1), (t+2), etc.

---

> > > ### Author Response · Authors · 2022-11-29
> > > **Thank You For Your Response**
> > >
> > > Thank you for taking time and providing additional feedback on our paper. Please see our response below to each of your questions.
> > >
> > > **R4-c) Thanks for the confidence intervals. The intervals for DT and LEAP overlap for the first three environments which makes me wonder if the results are really significant. Given that Atari is the only partially observable testbed that you used, the results are not very convincing that LEAP will generalize to more realistic settings. Was there a particular reason why you used fully observable setting in BabyAI?**
> > >
> > > Hi, we realized when we initially posted the standard deviations in Table 3, we posted the standard deviation of individual runs as opposed to the standard error of the estimated mean across environments. Using the standard error, the intervals are no longer intersecting: our performance is 378.9 +- 64.3 compared to DT’s 267.5  on Breakout, our performance is 19.6 +- 2.2 compared to DT’s 15.4 and on Pong our performance is 108.9 +- 1.6 compared to DT’s 106.1. We are happy to run further additional seeds of evaluation to make these confidence intervals even smaller and more precise. We also note that in Table 1 of the DT paper, all confidence intervals in Atari overlap that of a BC agent.
> > >
> > > The primary question we wish to study in this paper is how to construct an agent which may effectively plan in the environment (as opposed to constructing an effective RL agent).  While planning may be done in partially observed environments, effective planning is relatively easy in such settings as agent’s information about the environment is limited and it must replan frequently (i.e. simply planning 2 or 3 steps in the future may be close to optimal behavior).  In contrast, planning in fully observed environments is substantially more difficult (as the agent can stick to an optimal plan it makes) and there will be a substantial difference between the performance of an optimal planning agent and an agent that only plans 2 or 3 steps in future. Thus to increase the difficulty of the planning problem, we consider fully-observable BabyAI. Note that it is common to assume fully observable planning in robotics – such as in table-top rearrangement planning or task and motion planning.
> > >
> > > **R1) Thanks for the additional baselines. Could you please discuss in more detail why your method performs favorably compared to both model-based baselines? Why do these algorithms fail when LEAP is not failing? Could you also clarify if you are also using whole environment state for both of these methods?**
> > >
> > > Yes, all baseline methods receive the entire environment state. We believe the main reason in which our approach outperforms our compared to baselines is due to our iterative planning procedure – where we sequentially refine our planned trajectory for many iterations (and learn an energy function which enables us to refine trajectory over a long duration). In contrast, our baselines plan actions auto-regressively.
> > >
> > > **R4-b) It would help if you use the index consistently where (1) denotes the beginning of an episode, (2) next state, etc. or alternatively add (t) for current state like (t+1), (t+2), etc.**
> > >
> > > Thanks for the suggestion, we will update the symbols accordingly.

---

### Author Response · Authors · 2022-11-15
**General Response (1/2)**

We thank all reviewers for their careful reading and thorough feedback. We want to first address general concerns shared by several reviewers with further experiment results, including: model-based planning benchmark [gPMg, Vx64], stochastic setting performance [gPMg, MKux]. We then respond to specific questions and add additional **reviewer specific experiments** in the **individual responses** below. The revised contents are highlighted with red in the new version of submission.

# Additional Experiments

## Comparison with Model-based baselines

In response to the request for comparison between our method and model-based planning approaches from reviewer gPMg and Vx64, we have added a model-based planning baseline **PlaTe[1]** and a model-based reinforcement learning baseline **MOPO[2]**. The other suggested baseline **MuZero Unplugged[3]** by reviewer gPMg  is not included since their source code is not fully released. We collect the results in the table below, which is also updated in Table1 in the paper.
| Task | Env | LEAP | PlaTe | MOPO |
| ------------- | ------------- | ------------- | ------------- | ------------- |
| TP | GoToLocalS7N5 | **89.5%** | 42.5% | 87.0% |
| TP | GoToLocalS8N7 | **89.0%** | 45.0% | 81.5% |
| TP | GoToObjMazeS4 | **65.0%** | 35.5% | 60.0% |
| TP | GoToObjMazeS7 | **45.5%** | 30.5% | 30.5% |
| IC | PickUpLoc | **65.0%** | 7.5% | 43.5% |
| IC | GoToSeqS5R2 | **38.0%** | 25.0% | 30.0% |
| IC | GoToObjMazeS4R2Close | **55.5%** | 32.5% | 36.0% |

NOTE: TP stands for Trajectory Planning, and IC for Instruction Completion.
The results show that our method outperforms baselines in most environments, and performs comparably in the rest. Our better general performance against the both autoregressive methods further proves the advantage of multi-step planning against single-step.

1. PlaTe: Visually-Grounded Planning with Transformers in Procedural Tasks. Jiankai Sun, De-An Huang, Bo Lu, Yun-Hui Liu, Bolei Zhou, Animesh Garg.
2. Mopo: Model-based offline policy optimization. Yu T, Thomas G, Yu L, Ermon S, Zou JY, Levine S, Finn C, Ma T.
3. Online and Offline Reinforcement Learning by Planning with a Learned Model. Julian Schrittwieser, Thomas Hubert, Amol Mandhane, Mohammadamin Barekatain, Ioannis Antonoglou, David Silver.

## Stochasticity
In response to concerns in stochastic settings by reviewer gPMg and Vx64, here we discuss potential benefits of our method and present new experimental evidence.
Although [1], recommended by Reviewer MKux, shows that planning via sampling from the learnt policy conditioned on desired reward could lead to degenerate expected outcome due to the existence of stochastic factors, we believe our approach can be applied to stochastic settings since it plans by optimizing the learnt probabilistic model. Our training method trains the sequence model to assign low energies to trajectories with higher likelihood of success, as long as they have higher appearing frequency in the dataset. Therefore, in the stochastic environments, it constructs a sequence of actions that has the highest likelihood to reach the goal. When executing this plan in a stochastic environment, we may then choose to replan our sequence of actions after each actual action in the environment (to deal with stochasticity of the next state given an action). Note that this sequence of actions will be optimal at each timestep in the stochastic environment, as we always choose the action that has the maximum likelihood of reaching the final state.
Note that multi-step planning can potentially provide advantage over a simple policy to predict the next action in stochastic environments, as such policy simply assigns probability distribution to the immediate next step without the awareness of future step adjustments facing stochastic factors. To verify this conclusion,  we have designed a stochastic experiment in the BabyAI, where each turning action has a random chance to fail (details in Appendix B). In this setting, we find that our approach, where a sequence of actions in a stochastic environment is planned,  obtains a better performance than the single-step baseline DT. More extensive results and discussions are found in the Appendix B.

| Env | LEAP | DT |
| ------------- | ------------- | ------------- |
| GoToObjMazeS4 | **57.5%** | 30.8% |
| GoToObjMazeS7 | **33.3%** | 28.3% |

[1] You Can't Count on Luck: Why Decision Transformers Fail in Stochastic Environments. Paster, Keiran, Sheila McIlraith, and Jimmy Ba.

---

> ### Author Response · Authors · 2022-11-15
> **General Response (2/2)**
>
> # General Clarifications.
>
> ## Symbol Clarification.
> As pointed out by a few reviewers (gPMg, VX64), the input and output of the model are not clearly illustrated from the equations. Hence we redesign the annotations in the paper for better clarity.
> We want to first clarify here that the model inputs include both **context trajectory** $\tau_{ctx}$, defined as the concatenation of past few states and actions, and the future action sequence $a_{1:T}$ to optimize. The planning model takes in $\tau = (\tau_{ctx}, a_{1:T})$ and iteratively refine the $a_{1:T}$ component to achieve the lowest energy. Note that we adopt different contexts $\tau_{ctx}$ depending on the tasks. In BabyAI where the optimal strategy can be uniquely defined based on the current state, we define the context to be the current state $\tau_{ctx} = s_1$. On the other hand, we use past 25 states and actions in the atari environment as described in the Appendix A.4, as the current state - extracted from the current screenshot of the game - cannot fully describe the condition. Take the Breakout for example, the screenshot capturing the current position of the ball does not indicate whether the ball is rising or falling. Richer context in this case disambiguates the situation, which helps our model achieve a better result. The same strategy is used for baselines if applicable.
> The annotations and explanations above are updated in the main paper, Sec. 3
>
> ## Others
> We have uploaded our code to ensure reproducibility of our results.

---

### Decision · Program_Chairs · 2023-01-20

**Decision:**

Accept: poster

**Justification For Why Not Higher Score:**

While the authors managed to address many of the reviewers' concerns, some reservations still remain about the general applicability of the method.

**Justification For Why Not Lower Score:**

All of the reviewers agree that the paper should be accepted.

**Metareview: Summary, Strengths And Weaknesses:**

The paper presents an approach to using a masked language model trained on expert trajectories to generate sequences of actions that minimize a specified energy function. The method is tested on a subset of BabyAI and Atari tasks and compared to several baselines, and shows improved performance on the BabyAI tasks and comparable performance on Atari tasks. The paper also explores the use of the approach to solve compositional problems. Strengths of the paper include its novel approach to integrating language models with planning, and its ability to generate sequences of actions that minimize energy. The paper's weaknesses include its limited experimental evaluation and the lack of comparison to state-of-the-art methods, however many of these weaknesses were address during the discussion period, with all the reviewers suggesting that the paper is accepted.



**Note From Pc:**

if the above contains the word "oral" or "spotlight" please see: "oral" presentation means -> notable-top-5% and "spotlight" means -> notable-top-25%. As stated in our emails, we are disassociating presentation type from AC recommendations

**Summary Of Ac-Reviewer Meeting:**

I didn't hold the meeting as the reviewers engaged with the authors and updated their scores after my ping in a way that made this paper non-borderline.